# JUST-IN-TIME AND DISTRIBUTED TASK REPRESENTATIONS IN LANGUAGE MODELS

## ABSTRACT

Many of language models' impressive capabilities originate from their in-context learning: based on instructions or examples, they can infer and perform new tasks without weight updates. In this work, we investigate *when* representations for new tasks are formed in language models, and *how* these representations change over the course of context. We study two different task representations: those that are "transferrable"—vector representations that can transfer task contexts to another model instance, even without the full prompt—and simpler representations of high-level task categories. We show that transferrable task representations evolve in non-monotonic and sporadic ways, while task identity representations persist throughout the context. Specifically, transferrable task representations exhibit a two-fold locality. They successfully condense evidence when more examples are provided in the context. But this evidence accrual process exhibits strong *temporal* locality along the sequence dimension, coming online only at certain tokens—despite task identity being reliably decodable throughout the context. In some cases, transferrable task representations also show *semantic* locality, capturing a small task "scope" such as an independent subtask. Language models thus represent new tasks on the fly through both an inert, sustained sensitivity to the task and an active, just-in-time representation to support inference.

FIX

FIX

## 1 INTRODUCTION

Much of the excitement about large language models began with the discovery that they exhibit In-Context Learning (ICL; Brown et al., 2020): the emergent ability to learn tasks from few-shot examples in context. This discovery has led to a variety of works exploring the behavioral features of ICL (e.g. Sclar et al., 2024; Min et al., 2022). Other works have studied the dynamics of ICL, and how performance improves with increasing numbers of few-shot examples (Agarwal et al., 2024; Anil et al., 2024). The strong behavioral success of ICL led to substantial interest in understanding the mechanistic basis of these capabilities, leading to discoveries such as induction heads (e.g. Olsson et al., 2022), the core circuits responsible for learning from in-context examples (Cho et al., 2024; Bakalova et al., 2025), and how ICL implicitly refines a model of in-context evidence (e.g. Akyürek et al., 2022; Von Oswald et al., 2023).

FIX

Recently, several works have identified internal, vector-form task representations that can be extracted from a model's forward pass on a few-shot prompt (Todd et al., 2024; Hendel et al., 2023). These task representations not only capture general task information, but can be used to restore the appropriate task context during the model's forward pass on a zero-shot prompt. This transfer effect is observed by intervening with that representation at the appropriate place in the model's residual stream. Such intervention reinstates the task context and allows the model to perform the task without any explicit instructions or demonstrations. These "transferrable" task representations have been shown to exist across a variety of tasks and presentation formats, and even capture transferrable task knowledge across modalities (Davidson et al., 2025; Huang et al., 2024; Luo et al., 2024).

The discovery of transferrable task representations raises several intriguing questions about models' representations of new tasks. How and when are transferrable task representations formed throughout the context? How do they differ from other types of task representations that models may use? A simple, intuitive hypothesis is that models develop representations for new tasks gradually. Task

FIX

FIX

FIX

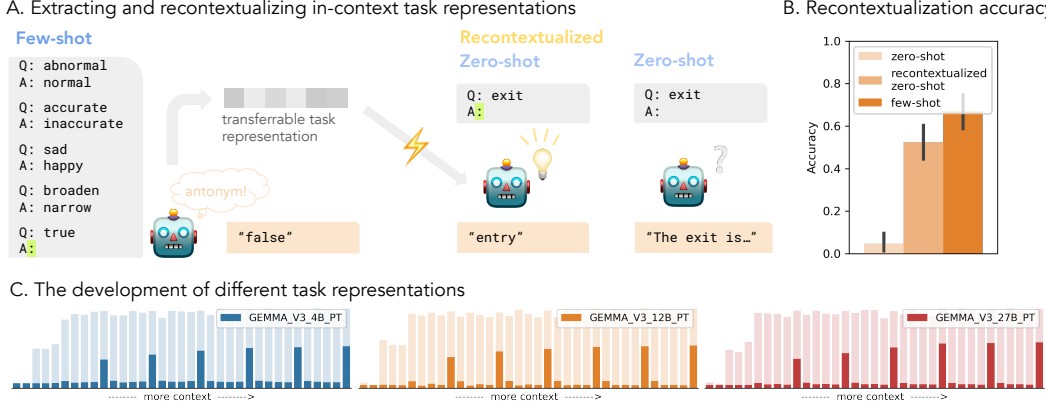

Figure 1: Understanding how task representations develop over context. **A**. A schematic of extracting transferrable task representations and restoring task contexts (via patching) in zero-shot settings. The highlighted tokens indicate the source and target for extracting and injecting task representations. **B**. Transferrable task representations restore task accuracy on zero-shot prompts. Results are aggregated over all models for simple tasks (see Appendix A). Error bars indicate the 95% CI over tasks. **C**. An overview of the development of different task representations over context. Solid bars: recontextualized zero-shot accuracy for task vectors extracted from different tokens. Transparent bars: task identity decoding accuracy from different token representations.

representations might reflect evidence accrual across tokens in the context and refine monotonically into more stable and robust representations. This view aligns with the behavioral findings that models perform better with more examples in-context (Agarwal et al., 2024; Anil et al., 2024).

We set out to investigate how the dynamics of ICL are reflected different types of task representations. FIX
Our findings suggest a nuanced picture:

- We find a stark contrast between two different task representations: an inert representation FIX
  of task identity is more continuously presented throughout the context, but transferrable task
  representations only activate sporadically at key tokens. FIX
- The fleeting but transferrable task representations can support longer generation. However,
  their ability to guide model behavior also decays across independent subtask contexts.
- Models rely on more distributed task representations in more complicated tasks that require FIX
  state tracking or chaining multiple subtasks together.
- Finally, models form distinct representations of a task when solving it independently vs. as FIX
  part of a broader context.

Overall, these results give us a window into language models' changing state when inferring and solving new tasks in context, but paint a complex and nuanced picture of the dynamics of this state. There are different types of task representations—identifiable vs. transferrable—that evolve over the context in distinct ways. The representations of tasks also depend on the task complexity and the surrounding context structure in which a task is embedded. These results may have implications for both the science of understanding models, and practical applications of mechanistic interpretability for analysis and safety.

## 2 RELATED WORK

Since the discovery that large language models exhibit emergent in-context learning (Brown et al., 2020), there has been substantial interest in investigating this capability and its mechanistic basis. From a behavioral perspective, many subsequent works have explored how ICL could develop from implicit meta-learning of data properties (Xie et al., 2022; Chan et al., 2022), and how this may relate to the broader set of language model capabilities (Chen et al., 2024; Lampinen et al., 2024).

Some of this work has focused on the surprising fragility of ICL to subtle prompt changes (e.g. Sclar et al., 2024); conversely, others have highlighted how ICL may be *overly* robust, allowing "learning" common tasks even if the labels are randomized (Min et al., 2022). One particularly relevant focus of behavioral work on ICL has been on the *dynamics* of in-context learning; for example, how adding many example shots can improve performance on difficult tasks, or even those discouraged in post-training (Agarwal et al., 2024; Anil et al., 2024).

From a mechanistic perspective, Olsson et al. (2022) showed in-context learning is supported by induction heads, and other work has studied how they might develop over training (Edelman et al., 2024; Singh et al., 2025). Extensive ablation has also elucidated core circuits supporting the aggregation of demonstrations from few-shot examples (Bakalova et al., 2025; Cho et al., 2024). Some recent work found that models may create transferrable task representations. These are representations that can be extracted from few-shot prompts and then injected (without the few-shot examples in context) to induce task performance. Hendel et al. (2023) demonstrated an instance of such representation: representations at intermediate layers of the last token in few-shot prompts can be injected to mimic the effect of a few-shot prompt. Concurrent work from Todd et al. (2024) identified "function vectors," which aggregate the effects of multiple attention heads to convey task information. Subsequent work has generalized these findings, exploring how function vectors can emerge from instructions (Davidson et al., 2025) and how task vectors capture task representations across modalities (Huang et al., 2024; Luo et al., 2024). Other works have explored how these representations emerge over training (Dong et al., 2025; Yang et al., 2025b; Yin & Steinhardt, 2025), revealed their limitations (Dong et al., 2025; Tikhonov et al., 2025), and extended the methods to more robustly restore task contexts in zero-shot settings (Li et al., 2024; Saglam et al., 2025). Building on these studies, our work characterizes and compares the dynamics of different task representations both *within* examples in individual tokens and *across* the context, as well as across different task types.

## 3 METHODS

**Tasks**  For our analyses, we built upon tasks from prior work on transferrable task representations (Hendel et al., 2023; Todd et al., 2024). The tasks we examine include a diverse set of natural language tasks (e.g., finding the antonym of a query word or translating an English word to French) and algorithmic tasks (e.g., counting or extracting a target word from a list of input words). In addition to these simple, single-token generation tasks from the previous literature, we also test a range of new tasks to explore model behavior in longer generation settings. These include: repeating a simple task three times (e.g. ANTONYM X 3 requires finding the antonyms of three input words), extracting multiple words from a query word list (e.g., choose both the first and the last word in the list), and reversing or shifting an entire word list. Finally, we also explore a set of "mixed-generation" tasks, where the model needs to infer and perform different tasks on each input item. See Appendix A for the full set of tasks.

There are 512 query-answer pairs for each task (except for two smaller datasets: COUNTRY-CAPITAL contains 197 samples, and PRODUCT-COMPANY contains 494 samples). These query-answer pairs are formatted into few-shot prompts with alternating "Q:" and "A:" turns, as shown in Figure 1A.

**Models**  In the main paper, we present results on the open-weight pre-trained 4B, 12B, and 27B Gemma V3 models (Team et al., 2025). In Appendix C, we also show that the main findings replicate on the 4B, 8B, and 14B Qwen3 models (Yang et al., 2025a).

**Decoding task identity**  We study a simple representation for new tasks given in-context evidence: whether token representations throughout the context contain robust task identity information. We trained simple linear decoders to predict the task category from the layer residual activations of different tokens. At each layer and token combination, we used 100 token instances for each task to train and test a task identity decoder. All decoders were trained for 20 epochs, with a batch size of 256 and a learning rate of 0.01. We report the decoding accuracy across the 25% held-out test representations.

**Extracting transferrable task representations**  We primarily investigate task vectors discovered in Hendel et al. (2023) as a window to study language models' transferrable task representations. In

Appendix D, we show that alternative extraction methods of transferrable task representations like function vectors show consistent results. We extract task vectors from few-shot prompts consisting of query-answer pairs and a test query, as shown in Figure 1A. Task vectors are the layer residual activations extracted from the last token before answer generation (in the example in Figure 1A, this corresponds to the highlighted colon token). Hendel et al. (2023) showed that task vectors can reinstate task performance on a different query even without any prior context. Specifically, when task vectors are patched onto (i.e., overwrite) the layer residual activations of the last token, they can recontextualize the model with the appropriate task context and enable the model to generate the task output without any prior few-shot examples.

We replicate and extend the procedure outlined in Hendel et al. (2023). For each model and task, we first search for the layer that best captures the task representation, using 50 queries from the dataset as the development set and in an 8-shot setting. As in prior work, we replace the real test queries with dummy queries sampled from the dataset to extract query-agnostic, general task representations. We searched among every 3 layers starting at layer 2 (0-indexed) for the 4B and 12B models (covering both the local-attention layers and global attention layers in Gemma V3 models; Team et al., 2025), and every 6 layers starting from layer 5 in the 27B model (covering the global attention layers). The layer that restores the highest task accuracy on zero-shot prompts in the development set is designated as the layer that best captures the representation for a given task. This best layer is subsequently used to extract task vectors and restore task contexts for the remainder, held-out queries in the dataset. Consistent with prior results, we generally find that task vectors extracted and patched at middle layers restore the highest task accuracy on zero-shot prompts, for all model sizes.

**Evaluating task transfer**   We compare the average accuracy of the sampled responses across three settings: standard zero-shot, recontextualized zero-shot (with task vector intervention), and few-shot (with examples in context). For simplicity, responses for all tasks are graded by exact string matches against the ground-truth answer. This underestimates the model performance in some tasks (e.g. for antonym and translation tasks), but we use the same grading scheme across all settings and compare relative performances. For longer-generation and mixed-generation tasks, we evaluate each of the multiple outputs separately by exact match (e.g., in ANTONYM X 3, we compare each of the three output words with the correct answer), and report the average accuracy across all output units.

**Examining the dynamics of transferrable task representations**   Once we determine the best layer for each task using the last colon token, we evaluate how well the colon token representations condense information from multiple examples in a prompt. We do this by extracting task vectors at the colon token in different $k$-shot prompts and patching onto zero-shot prompts, then comparing the recontextualized zero-shot accuracy. We repeat this analysis for $k$ in 0, 1, 2, 4, 8, 16, 32. In an earlier experiment on a subset of the tasks, we also experimented with allowing the best layer to vary depending on $k$, but found very similar results overall. We show these layer search results across different $k$'s in Figure 2A, but otherwise focus on results from reusing the best layer from the layer search on 8-shot prompts for other $k$'s.

We then repeat this intervention on other tokens to understand when transferrable task representations form. We extracted layer residual token activations for other format tokens in the context, including the "Q", the ":" following "Q", the "A", and the new-line token before "A". We patched these token activations onto the corresponding token in the zero-shot prompt at the same layer. For each of the non-colon tokens, we repeat the search for the layer that best captures task representations. All token representations are evaluated on the extent to which they restore task accuracy on zero-shot prompts.

Lastly, we also explore a set of cross-token analyses to investigate the extent to which different tokens share representational or functional roles (see main text).

**Characterizing the state of task representations.**   We use specific terminology to describe the representational dynamics of the model's internal state, distinct from the methodological operations (e.g., activation patching) used to probe them. By recontextualization, we refer to the phenomena that transferrable task representations reinstate task contexts and enable inference in a zero-shot setting. When discussing temporal properties, we refer to the distribution of representations along the sequence dimension. Finally, we use task scope to capture the semantic extent of the inference behavior enabled by recontextualization.

## 4 RESULTS

How do language models represent new tasks in-context? We use the transferability of task contexts    FIX
to understand when models form transferrable task representations, as opposed to maintaining a    NEW
simple representation of task identity. We first show that transferrable task representations like    FIX
task vectors indeed aggregate in-context evidence. We then show that this evidence accrual process
happens in a sporadic way, with transferrable task representations only forming at certain tokens
(Figure 1C, also see Figure 2A). This is in strong contrast to the persistence of representation for task
identity across tokens in the context (Figure 2B). We also find that, in different settings, transferrable    FIX
task representations can support generation for longer tasks or only support a minimal "task scope."
Below, we discuss these findings in more detail.

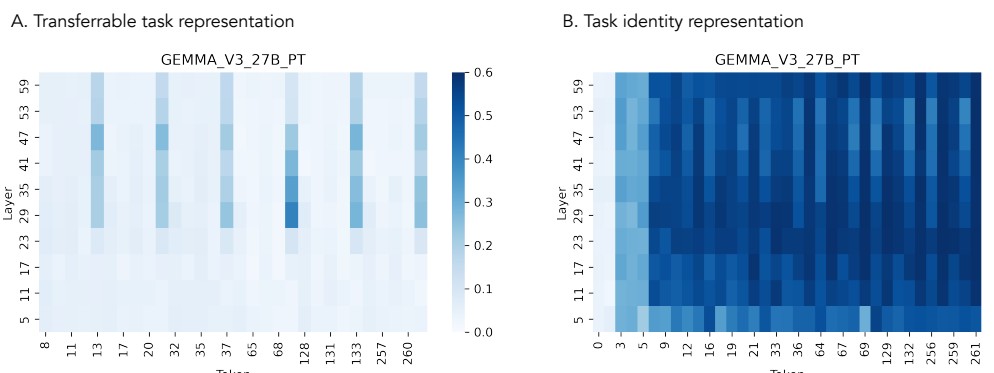

Figure 2: Transferrable task representations activate sporadically at key tokens, but task identity
representations persist throughout the context. **A.** Recontextualization accuracy when each token
representation is used to restore task contexts in zero-shot settings. **B**. Task identity decoding accuracy
(among 14 tasks) for token representations at different layers and positions. This figure plots aligned
sequences across different samples and tasks; since exact positions differ depending on the sample,
the indices shown in the labels are approximate. See results for other models in Figure S4.

### 4.1 TRANSFERRABLE TASK REPRESENTATIONS ACCRUE EVIDENCE

**Transferrable task representations reflect evidence accrual.**    Consistent with the behavioral gain
from including more examples in-context (e.g., Anil et al., 2024), we confirm that transferrable task    FIX
representations also reflect increased task certainty with increased context (Figure 3A). Task vectors
extracted following more examples are better at restoring task performance in zero-shot settings, such
that the ratio between the recontextualized zero-shot accuracy and few-shot accuracy stays relatively
stable across the number of examples. This suggests that language models condense information
from multiple examples and form better task representations, even though the task representations
extracted are fairly local (i.e., from a single token in the few-shot prompts).

**... but not for all types of tasks.**    However, we did not observe strong evidence accrual in 2 out of
the 14 simple tasks (Figure 3A, hard-to-transfer tasks). These two tasks are COUNT_COLOR_IN_3
and COUNT_FRUIT_IN_3. For all three model sizes, task vectors extracted from 32-shot prompts
recovered below 50% of the few-shot accuracy with 32 examples in the context. In other words, local
task representations for these tasks were not able to take advantage of more examples for task transfer
to zero-shot settings, even though models improved substantially at solving the task when given more
examples in the prompt. Interestingly, the same counting tasks were hard-to-transfer for Qwen3
models as well (see Figure S7A). One possibility is that these tasks require more state-tracking,
which may necessitate additional inference processes that the models do not condense into local task
representations. Alternatively, these inference processes cannot be effectively re-activated by the
injection of the extracted task representations.

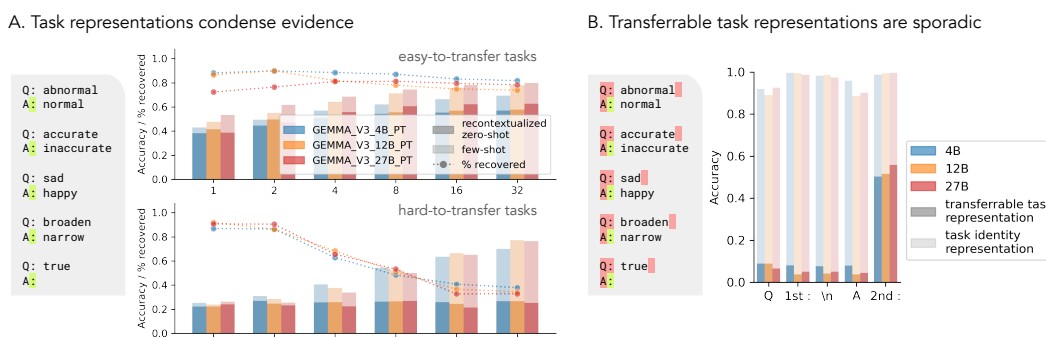

Figure 3: Sporadic & inconsistent evidence accrual in language models. **A**. Task vectors extracted from the last colon token in each example capture evidence accrual on most tasks (12 out of 14). However, on two "hard-to-transfer" tasks, task vectors do not capture this evidence accumulation, even though the models (behaviorally) do learn from more examples. The solid bars indicate recontextualized zero-shot accuracy (via task vectors), and light bars in the background indicate few-shot accuracy (without task vectors). The dotted lines indicate the ratio of the recontextualized zero-shot accuracy against few-shot accuracy. **B**. Most other format tokens in the context do not robustly form transferrable task representations that support recontextualization on zero-shot, but task identity is reliably decodable in their residual activations. Here, we report the task identity decoding accuracy at the mode best layer at which transferrable task representations form in the second ":" token. See the main text for more details.

**How evidence accrual leads transferrable task representations to converge.** As models appear to successfully accrue evidence in these highly local representations, we sought to understand how the task representations themselves changed over more examples (Figure 4A). We look at the task vectors extracted from the mode best layer across different tasks. This is to control for magnitude differences of the residual activations across layers and make a fair comparison. Although the best layer for transferrable task representations sometimes differ across tasks, the best layers tend to reside in the middle layer range across all model sizes, consistent with prior findings (Hendel et al., 2023).    FIX

In Gemma V3 models, as we increase the number of examples, we generally found reduced variance among task vectors extracted from different k-shot prompts. This can signal that in-context task representations tend to denoise or converge to more stable representations as models gain evidence. The magnitude (L2-norm) of the task vectors also tends to decrease over time. However, for both the variance and magnitude, there were considerable differences between tasks and models. Some tasks seem to converge to stable representations faster (i.e., with fewer examples; see also a visualization of the representational trajectories in Figure S2). For certain tasks, the magnitude of the task representations first increases then decreases given more examples. We also note that both the variance and magnitude of task vectors tend to increase per more examples in some Qwen3 models (Figure S8), suggesting that different models may develop different strategies for refining task representations.

## 4.2 DIFFERENT TASK REPRESENTATIONS EXHIBIT DISTINCT TEMPORAL PROFILES

The analyses above confirm that transferrable task representations benefit from increasing in-context evidence and converge to better representations. To understand the full temporal profile of this accrual process, we repeated the task vector intervention on *other* tokens in the prompt, which revealed that these representations do not strengthen monotonically. We show that this temporal locality is unique to transferrable task representations, as representations for task identity are widespread.    NEW

**Transferrable task representations are not found in most tokens.** We tested whether extracted representations from other format tokens can also restore the corresponding task contexts. These include "Q", the ":" following "Q", "A", and the new-line token before "A", which are all shared across examples, tasks, and contexts. As before, we patched the activations at the same layer, but onto

A. Variance and direction of task representation updates

B. t-SNE projection of all task representations

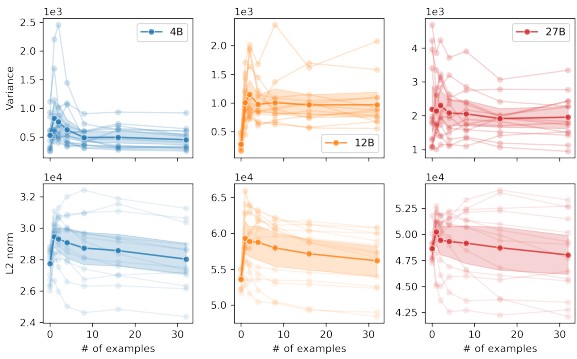

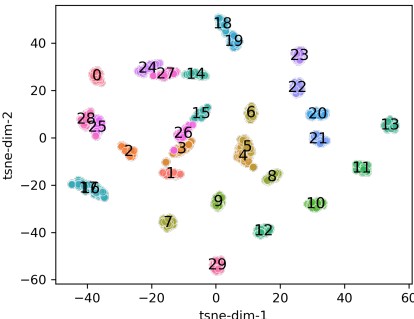

Figure 4: Analyses of extracted task representations in Gemma V3 models. **A**. The extracted task vectors (at the last colon token) tend to decrease in both variance and magnitude with more examples, exhibiting a general tendency to condense evidence and converge onto stable task representations. The solid line shows the average across tasks. The transparent lines show the individual tasks. **B**. Extracted task vectors from the 27B model form distinct clusters. The numbers label the centroid for each task (see legend and results for other models in Figure S5). Task vectors are similar but distinguishable when a task is evaluated independently vs. embedded within a larger task structure. For example, representations for ANTONYM (0), ANTONYM X 3 (14), and where ANTONYM appears as a first task in a mixed-generation task chain (24&27) are close but distinct.

the corresponding format token instead of the last colon token in zero-shot prompts. As shown in Figure 3B, transferrable task representations generally do not form in the residual activations across layers in these tokens. This is true across the number of examples provided in the prompt, leading to the developmental trajectory of transferrable task representations shown in Figure 1C.

We observed nearly zero recontextualized zero-shot accuracy for all these tokens in most tasks, except some restoration success in PRODUCT-COMPANY, COLOR_V_ANIMAL_3, CHOOSE_FIRST_OF_5, and the longer-generation tasks discussed below (see Figure S3). In Qwen3 models, there also appear to be some restoration successes with representations extracted from the "Q" token (see Figure S7B and Figure S11). These partial successes are likely driven by the fact that the first answer token sometimes match the first input token across a subset of the tasks. In general, it seems that an effective, transferrable task representation in language models only forms sparingly; in few-shot settings, this often means a just-in-time task representation at the token before answer generation.

**... but a robust task identity signal persists throughout the context.** Intriguingly, however, task identity is almost perfectly decodable in the representations extracted from *all* the different format tokens, even though the formats are shared across all tasks. We report the decoder accuracy from the layer with the best transferrable task representations in Figure 3B, but found high task identity decoding accuracy across most tokens throughout the context (Figure 2B). The decodability success may be partly due to vocabulary differences between some tasks, but we show that decoding accuracy remains high even for a restricted subset of tasks with shared vocabulary (Figure S4C). Both task identity and task transferability do not occur until at least one full example is presented, but accurate task identity representations form much earlier than transferrable task representations (Figure 1C). This suggests that the model is generally *task-sensitive*, but instantiates *transferrable* task representations only at particular timepoints in the context.

NEW

**Non-trivial cross-token representational and functional transfer.** We primarily observe task identity representations in all tokens and transferrable task representations in the key pre-answer colon tokens. However, we note that these two types of representations are not entirely binary. We performed a set of cross-token task vector transfer experiments: both using representations from non-key tokens in few-shot prompts to intervene on the key colon token in zero-shot prompts, and using representations from the key colon token to intervene on non-key tokens. More details are

presented in Appendix E. In both cases, we observe some non-trivial success of task recontextualiza-  NEW
tion. For example, representations extracted from the colon following the "Q" token (prior to query
presentation) can partially restore task contexts. In Qwen3 models, we also find that non-key tokens
can functionally participate in task recontextualization when injected with effective transferrable task
representations. These results suggest that task identity signals and transferrable task knowledge may
co-exist in token representations. Furthermore, tokens that do not usually form transferrable task
representations may nonetheless preserve a functional role to establish task contexts.  NEW

We then further explore the overlap between identifiable representations across tokens. We show that
task identity regressions fit on non-transferrable representations generalize to transferrable ones, that
there is partial overlap between the identifiable-non-transferrable subspaces and the transferrable
ones, and that the degree of generalization and overlap is modulated by the same task features as in
our other experiments. These results reinforce our claim that these different task representations have
distinct dynamics and can coexist, but show additional nuance in their partial overlap.

### 4.3 TRANSFERRABLE TASK REPRESENTATIONS CAPTURE VARIANT SCOPE LOCALITY

We have seen evidence that transferrable task representations tend to be temporally local. That leads
to the question of whether they have a lasting effect over generation. That is, are the restored task
contexts in the zero-shot forward pass also fleeting in nature? To study the task scope supported  FIX
by these representations, we tested to what extent restored contexts can support longer generation
beyond the first token. Building on the simple tasks from prior work, we evaluated models on a set
of longer-generation and mixed-generation tasks, including repeating a simple task multiple times
on different input words, list-level tasks that operate over multiple words, and inferring/performing
different tasks on different words (see Methods and Appendix A).

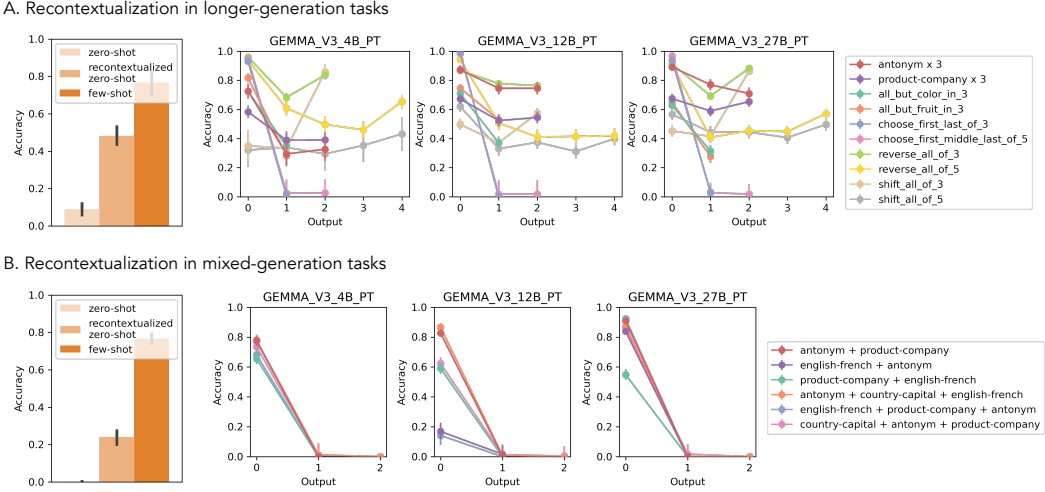

Figure 5: Reinstantiated task contexts in longer- and mixed-generation tasks often decay over
generation, especially for tasks that can be decomposed into semantically-independent subtasks. This
suggests a tendency for models to only activate transferrable representations for small task scopes.
**A**. Bar plot: recontextualized zero-shot accuracy compared to zero-shot and 8-shot accuracy on
longer-generation tasks; accuracies within each task are averaged across output units. Line plots:
recontextualization accuracy for each output unit, conditioned on sequences where models generated
full correct responses with eight examples in-context. An output unit usually corresponds to a single
word and is occasionally a short phrase (e.g. the capital of a country). **B**. Visualization as in A, but
for mixed-generation tasks.

**Transferrable task representations sometimes only support limited task scopes.** In these
experiments, we find further evidence on the semantic locality of transferrable task representations.

Overall, the recontextualized zero-shot accuracy of tasks that require longer and mixed answers is substantially lower than that in tasks that require shorter answers (Figure 5, bar plots; also see Figure S1). Across a range of tasks, we find that the recontextualized zero-shot accuracy decreases over longer generation (Figure 5, line plots; see similar results on Qwen3 models in Figure S13), suggesting that the restored task contexts sometimes "fade" over longer generation.

We notice a stark contrast between different tasks. In some longer-generation tasks that repeat the same task multiple times or require list-level operations, the reinstantiated task contexts (from task vectors extracted in the key colon token) successfully supported continued generation. In other tasks, however, this intervention only supported generating the first output. This effect is very pronounced in the mixed-generation tasks that combine multiple distinct subtasks. In these cases, all models form strong local task representations that only encapsulate the first subtask. This suggests that transferrable task representations at the key colon token can capture variant "task scopes", which tend to be limited in cases where the subtasks have distinct semantic boundaries.

We confirm that models defer representing later subtasks using additional interventions (see Appendix F). Indeed, we find that transferrable subtask representations in mixed-generation tasks form at the comma tokens prior to subtask answer generation. However, intervening comma tokens alone does not seem to recover the full subtask inference ability. We posit that the full subtask representation is distributed across colon and comma tokens. Interestingly, the extracted task representations from the colon tokens in the same simple task can be distinct when it appears independently or as a first task in a multi-task context (Figure 4B), even though they support generating the same responses. This may in-part reflect vocabulary differences in later subtasks, but potentially also information relevant for later subtasks to be activated, even though these representation alone are not sufficient to restore execution of later subtasks.

## 5 DISCUSSION

We sought to understand the dynamics of in-context task representations that support language models' successful learning of new tasks. We evaluated when in the context we can extract transferrable task representations that restore task contexts in zero-shot settings, and when we can extract robust representations of task identity. Our results show that, transferrable task representations only sporadically activate, even though they accrue evidence from multiple examples. However, models maintain a strong general sensitivity to high-level task differences that persists throughout the context. We find cases where models do not form a global task representation at particular token sites, such as tasks involving state tracking and tasks combining distinct subtasks together. We also find nuances in the distinction between the two types of task representations, where the representational and functional roles of different tokens exhibit non-trivial overlap in certain settings.

In general, our results complicate the intuitive picture that language models smoothly and gradually refine task representations during in-context learning (ICL). Models form task representations at different levels and different rates. Across tokens and examples in the context, the same task state is not sustained, but can fade and reactivate across different tokens. These high-level changes in models' representational states unite a few observations from prior work (Bakalova et al., 2025; Cho et al., 2024; Dong et al., 2025). Key tokens prior to answer generation help aggregate in-context examples but do not all participate in prediction circuits, as the effective task representations they once generate are deactivated and reactivated later. The convergence of transferrable task representations may result from the contextualization subcircuit identified in (Bakalova et al., 2025), such that later demonstrations contribute more to the test query as seen in (Cho et al., 2024). Tokens seemingly outside of a core ICL circuit may nonetheless restore task contexts when given the right information, potentially contributing to the bypass mechanisms noted in (Cho et al., 2024).

One trend that arose from these investigations is that language models do not seem to condense task information into local representations in all cases. We find that language models construct token-level task representations that flexibly capture more broad task contexts and longer outputs. But in some cases, models elect to form sharp local contexts only for small task units and offload task

representations across multiple tokens. This flexibility across different types of longer-generation tasks adds additional nuance to the distributed-ness of effective task vectors across multiple tokens similarly noted in Dong et al. (2025) and Tikhonov et al. (2025). This flexibility may also relate to the success many works have observed on extracting transferrable task representations in broad settings, such as following instructions or images (Davidson et al., 2025; Huang et al., 2024; Luo et al., 2024), or capturing information for multiple possible task outcomes (Xiong et al., 2024).  FIX

For some tasks that require more complicated computation such intermediate-state tracking, successful inference may need to rely on not only cross-token but cross-layer representations (e.g., as shown in Ameisen et al., 2025). In these cases, the effective restoration of the computation process may also require intervening multiple layers during the forward pass. It is also possible that by overriding token activations at intermediate layers with task vectors, the models have lost task state information formed in earlier layers. Here, we show that zero-shot recontextualization remains challenging in these tasks  NEW
even through additive injection such as function vectors (Todd et al., 2024, see Appendix D). But  FIX
more advanced methods that restore task states by intervening multiple layers may be more successful at restoring model task states in these cases and elicit different temporal dynamics (Li et al., 2024; Saglam et al., 2025). We also show some evidence that different models may differ in the extent to which they form local or distributed representations in different tasks.

An intriguing direction for future work is to study the mechanistic bases for the strong temporal and scope locality we observed in models' in-context task representations. One possibility may be that the residual stream is more stable and easier to learn from during training. This may encourage the model to rely more on the residual stream to condense contextual task representations rather than relying on the more expensive attention operations. These learning dynamics may drive models to conform with an implicit normative consideration to not instantiate task contexts until needed and instantiate just the right scope to avoid capacity waste. Some of these features may even be exclusive to models pre-trained on natural languages (e.g. Yang et al., 2025b).  NEW

Taken together, our findings offer some useful insights for interpretability and interactions with language models at large. For the science of understanding language models, comparing between more passive and more active, transferrable levels of representation offers a useful framework for probing representations of different knowledge or concepts. Practically, the variant temporal and semantic locality of different task representations means that interactions such as prompting or model steering should not assume a stable, continuous task state. Rather, effective control may require strategic re-instantiation of the fleeting representations at critical subtask boundaries.

**Limitation**   We note a few important limitations of our work. First, we primarily rely on single-  FIX
token, single-layer intervention methods to extract transferrable task representations. This means that our conclusions and speculations are bounded by the effectiveness of these methods. As we discussed earlier, representations for some task contexts may be more distributed, either across tokens and/or across model layers. It would be important to confirm if similar dynamics are observed in less-constrained methods such as a multi-layer recontextualization (Li et al., 2024; Saglam et al., 2025). Second, we mostly explored relatively simple tasks, including when we investigated longer-generation tasks. It's possible that many of the dynamics we observe here would not generalize to settings with naturalistic languages, especially when the tasks are not cleanly decomposable and a single, semantically-independent task unit is hard to define.

**Conclusion**   We investigated how the dynamics of in-context learning are reflected in the development of language models' internal task representations. Our results suggest that language models do not smoothly refine a global task state in-context. While general task sensitivity persists throughout  FIX
context, models construct a more active, transferrable task representation in a "just-in-time" fashion. These contrasting levels of task representations provide new insight into the models' state of inferring and performing tasks based on new evidence.

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

## A TASKS

Table 1: Simple/shorter-answer tasks. See Todd et al. (2024) for more details for the first nine tasks.

| Task Name | Example |
| --- | --- |
| ANTONYM | Q: true A: false |
| COUNTRY-CAPITAL | Q: Germany A: Berlin |
| ENGLISH-FRENCH | Q: queens A: reines |
| PRODUCT-COMPANY | Q: Windows XP A: Microsoft |
| COLOR_V_ANIMAL_3 | Q: blue, dolphin, swan A: blue |
| FRUIT_V_ANIMAL_3 | Q: lime, parrot, buffalo A: lime |
| CHOOSE_FIRST_OF_5 | Q: envelope, pasta, cake, toucan, create A: envelope |
| CHOOSE_MIDDLE_OF_5 | Q: candy, charismatic, laptop, realize, eel A: laptop |
| CHOOSE_LAST_OF_5 | Q: affable, believe, carefree, zoom, moray A: moray |
| WORD_LENGTH | Q: negotiate A: 9 |
| COUNT_COLOR_IN_3 | Q: snake, gold, indigo A: two |
| COUNT_FRUIT_IN_3 | Q: lime, newt, bunny A: one |
| POSITION_OF_COLOR_IN_3 | Q: monkey, oryx, white A: third |
| POSITION_OF_FRUIT_IN_3 | Q: pear, coyote, capybara A: first |

Table 2: Longer-generation tasks.

| Task Name | Example |
| --- | --- |
| ANTONYM X 3 | Q: fall, everybody, intact A: rise, nobody, broken |
| PRODUCT-COMPANY X 3 | Q: iWork, Windows NT 3.5, OS X Yosemite A: Apple, Microsoft, Apple |
| ALL_BUT_COLOR_IN_3 | Q: cat, black, pelican A: cat, pelican |
| ALL_BUT_FRUIT_IN_3 | Q: grape, butterfly, llama A: butterfly, llama |
| CHOOSE_FIRST_LAST_OF_3 | Q: white, house, wallet A: white, wallet |
| CHOOSE_FIRST_MIDDLE_LAST_OF_5 | Q: dolphin, beyond, curtain, pillow, intuitive A: dolphin, curtain, intuitive |
| REVERSE_ALL_OF_3 | Q: donut, sad, who A: who, sad, donut |
| REVERSE_ALL_OF_5 | Q: she, honest, out, test, frog A: frog, test, out, honest, she |
| SHIFT_ALL_OF_3 | Q: piano, cougar, jackfruit A: cougar, jackfruit, piano |
| SHIFT_ALL_OF_5 | Q: agreeable, flamingo, short, around, jovial A: flamingo, short, around, jovial, agreeable |

Table 3: Mixed-generation tasks.

| Task Name | Example |
|---|---|
| ANTONYM + PRODUCT-COMPANY | Q: opponent, iDisk A: ally, Apple |
| ENGLISH-FRENCH + ANTONYM | Q: liberal, continue A: libéral, stop |
| PRODUCT-COMPANY + ENGLISH-FRENCH | Q: Alfa Romeo MiTo, mask A: Fiat, masque |
| ANTONYM + COUNTRY-CAPITAL + ENGLISH-FRENCH | Q: upper, Greece, artists A: lower, Athens, artistes |
| ENGLISH-FRENCH + PRODUCT-COMPANY + ANTONYM | Q: system, Lancia Flavia, unlucky A: système, Fiat, lucky |
| COUNTRY-CAPITAL + ANTONYM + PRODUCT-COMPANY | Q: Gambia, heavy, Game & Watch A: Banjul, light, Nintendo |

## B ADDITIONAL GEMMA3 RESULTS

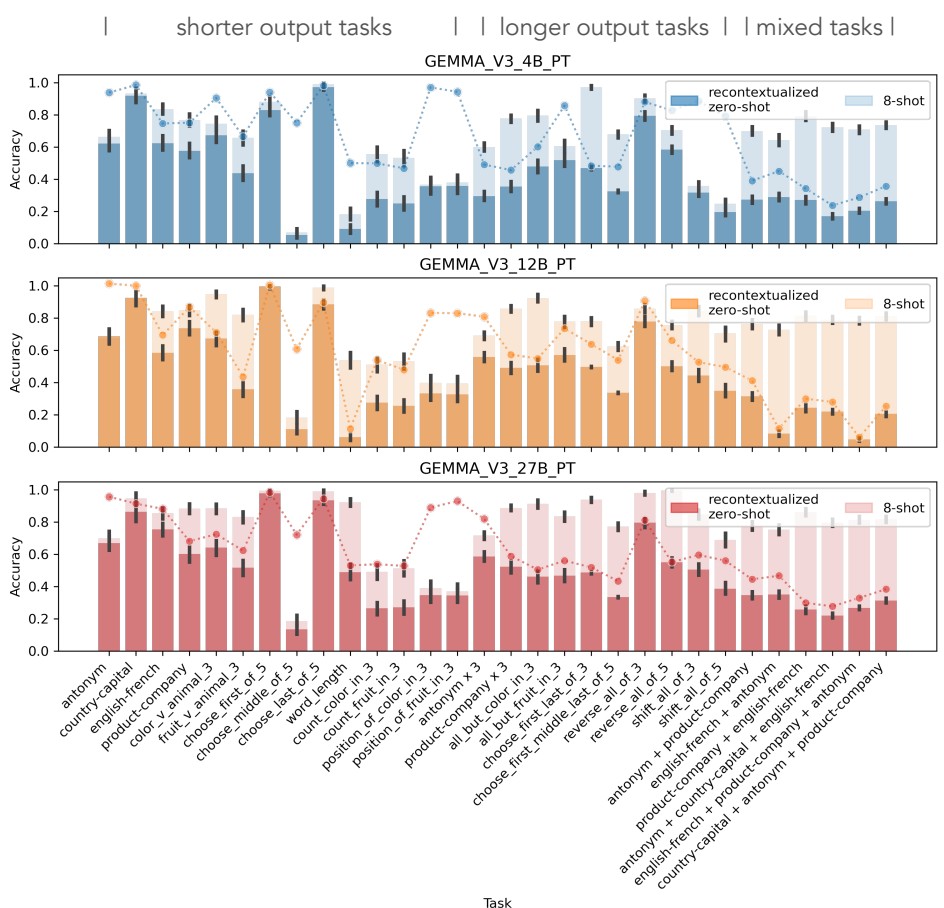

Figure S1: Recontextualization accuracy for all tasks. Task vectors extracted from 8-shot prompts are used to reinstantiate task contexts in zero-shot settings. The dotted line indicates the ratio between recontextualized zero-shot accuracy and 8-shot accuracy.

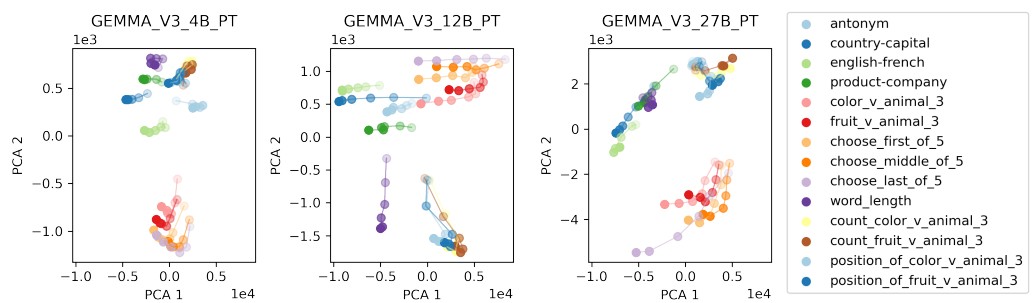

Figure S2: Developmental trajectory of task representations over shots. Task vectors are the token activations of the colon token prior to answer generation. We visualize task vectors sourced from the mode best layer across tasks at which task contexts are best restored in a zero-shot setting. Representations for each task are first averaged across samples with the same number of examples in the prompt.

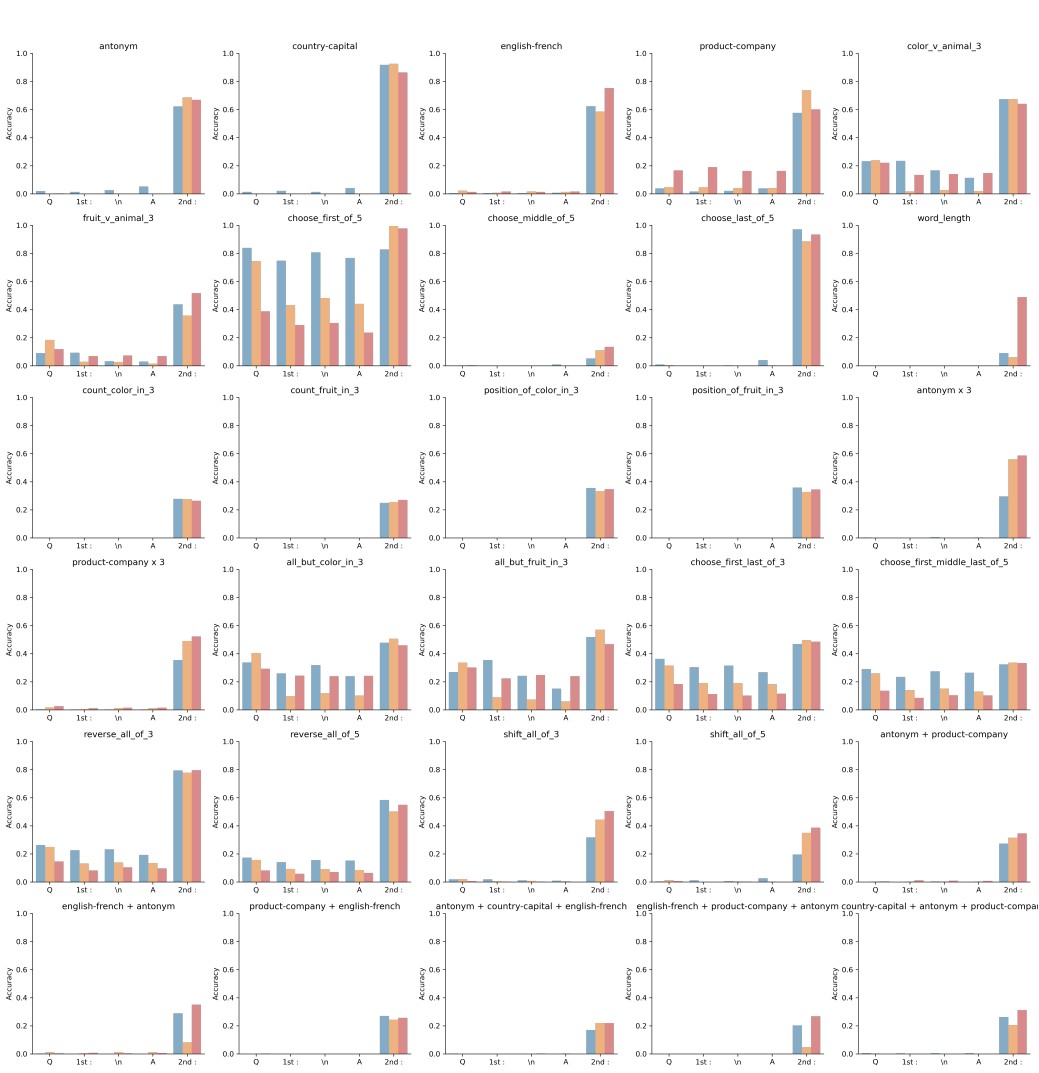

Figure S3: Recontextualized zero-shot accuracy from different format tokens in the prompt in different tasks. The colors indicate different model sizes: blue=GEMMA_V3_4B_PT, yellow=GEMMA_V3_12B_PT, red=GEMMA_V3_27B_PT.

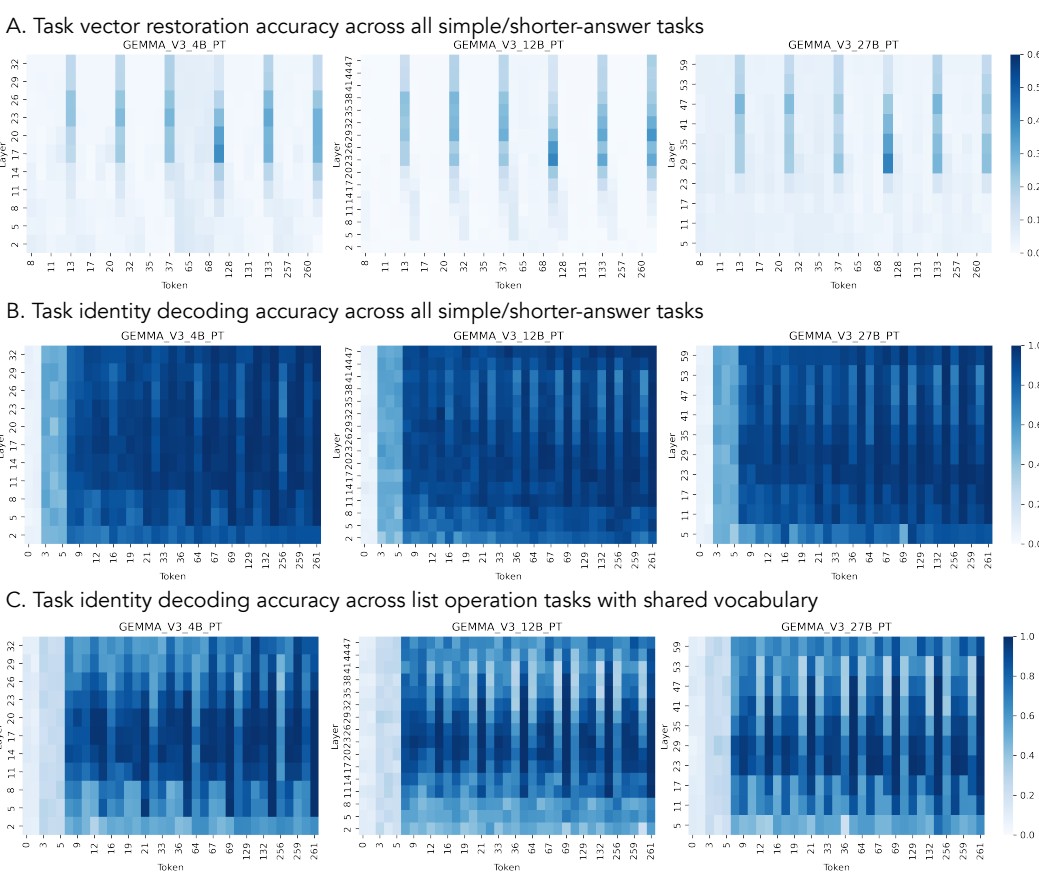

Figure S4: The development of different task representations over layers and context. **A**. Recontextualized accuracy when each token representation is used to restore task context in zero-shot settings. **B**. Task identity decoding accuracy across all 14 simple/short-answer tasks in Table 1. **C**. Task identity decoding accuracy across 9 list operation tasks with shared vocabulary. This subset of tasks includes: CHOOSE_FIRST_OF_5, CHOOSE_MIDDLE_OF_5, CHOOSE_LAST_OF_5, CHOOSE_FIRST_LAST_OF_3, CHOOSE_FIRST_MIDDLE_LAST_OF_5, REVERSE_ALL_OF_3, REVERSE_ALL_OF_5, SHIFT_ALL_OF_3, SHIFT_ALL_OF_5.

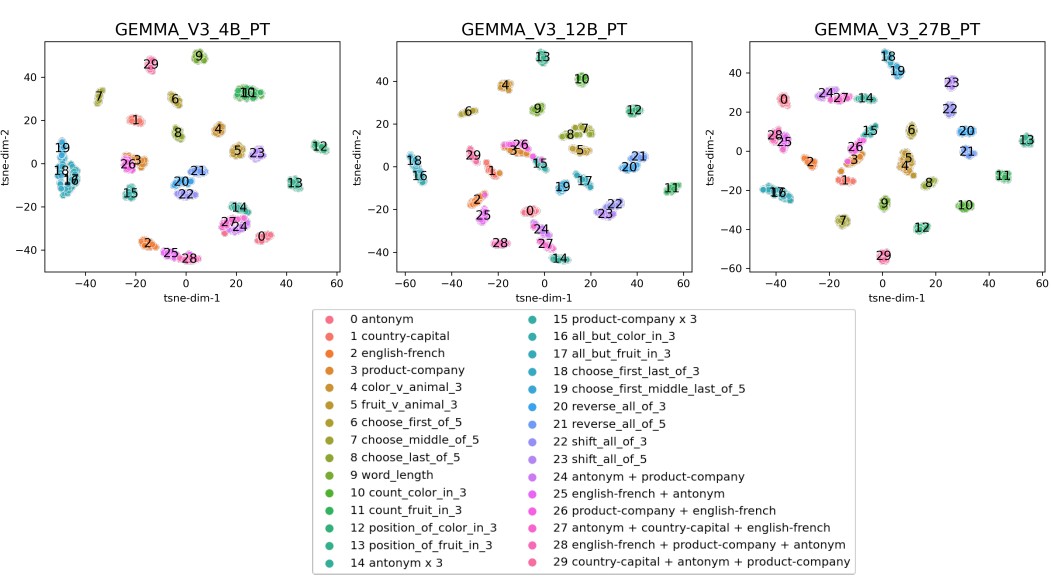

Figure S5: T-SNE projection of all task vectors across models and tasks.

## C  RESULTS ON QWEN3 MODELS

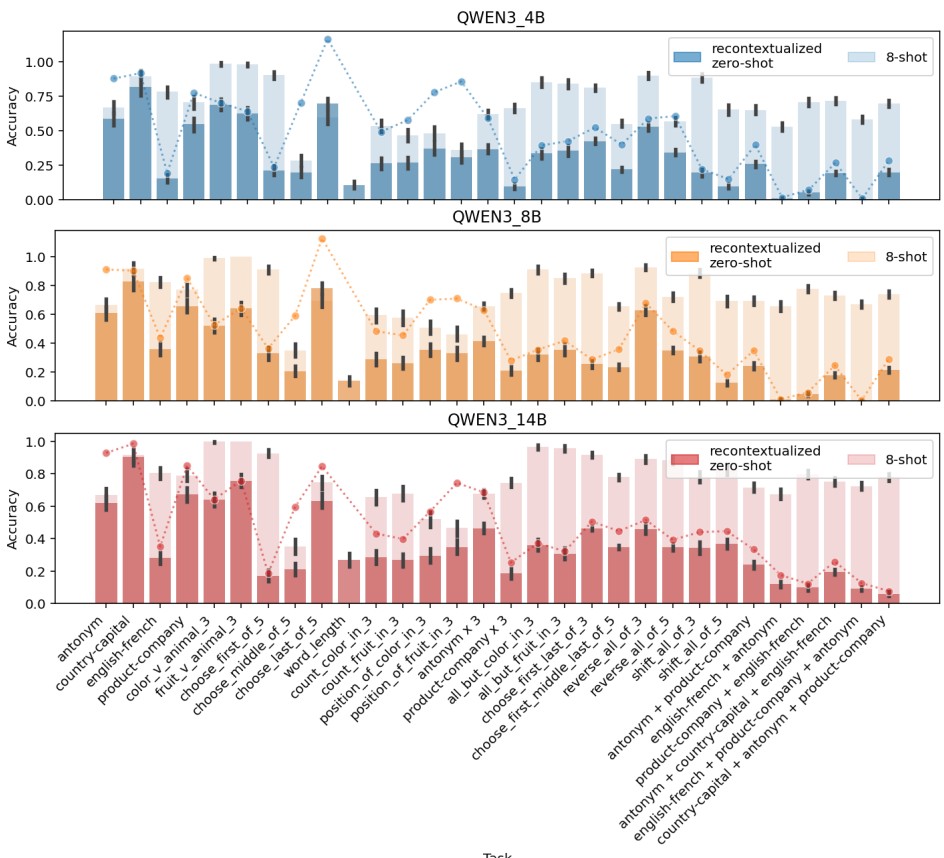

Figure S6: Recontextualization accuracy from Qwen3 models for all tasks. Visualization as in Figure S1.

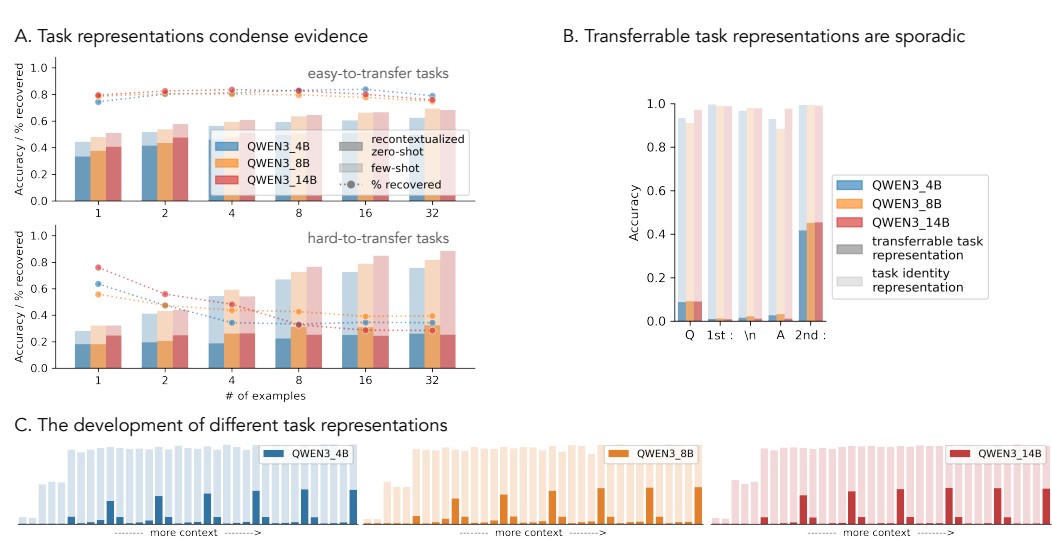

Figure S7: In Qwen3 models, transferrable task representations also condense evidence and activate sporadically at specific tokens. Visualization as in Figures 3A, B and Figure 1C. The hard-to-transfer tasks in Qwen3 models include ENGLISH-FRENCH, CHOOSE_FIRST_OF_5, COUNT_COLOR_V_ANIMAL_3, and COUNT_FRUIT_V_ANIMAL_3. These are the tasks where task vectors extracted from 32-shot prompts failed to recover more than 50% of the 32-shot accuracy on 0-shot prompts, across all model sizes.

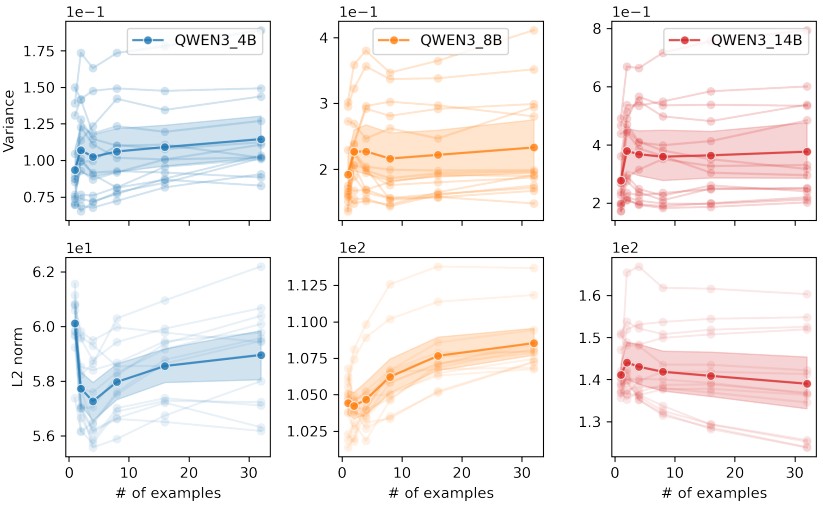

Figure S8: Variance and magnitude changes across task vectors extracted following different number of examples. Visualization as in Figure 4A.

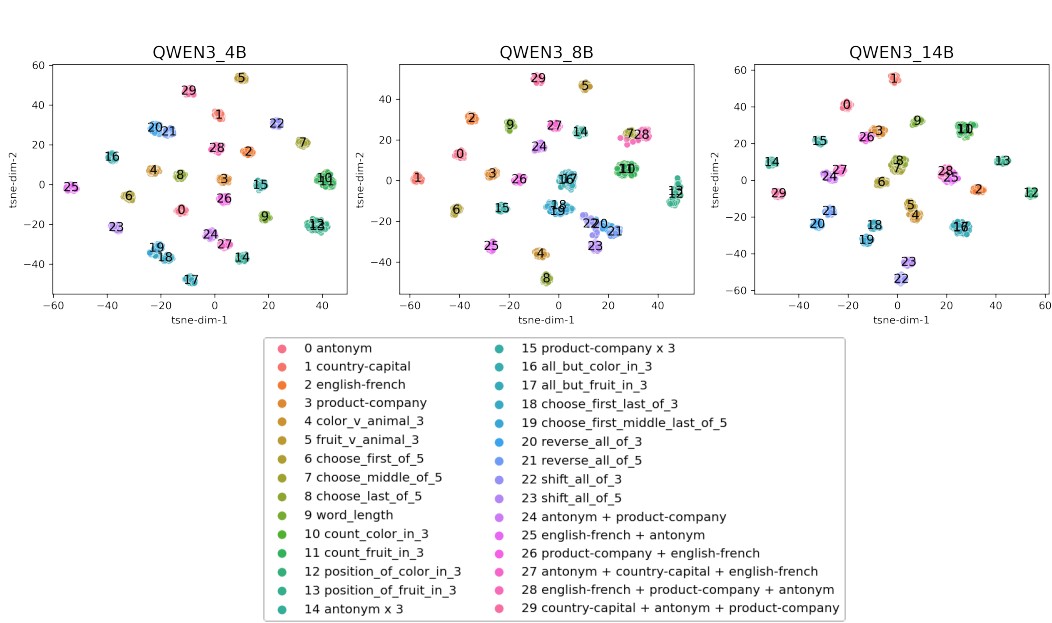

Figure S9: T-SNE task representations across all models and tasks.

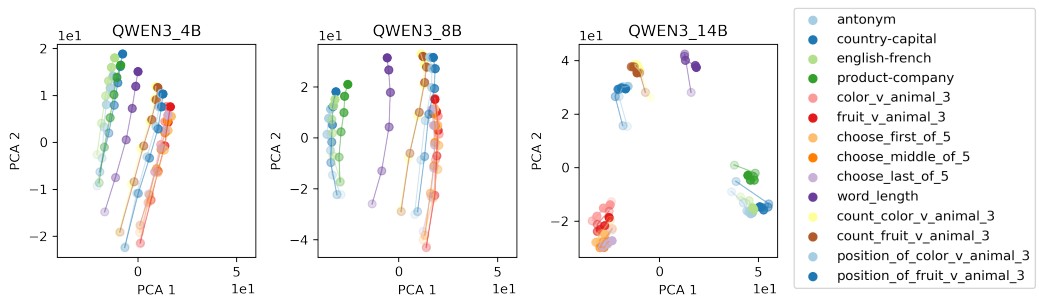

Figure S10: Developmental trajectory of task representations over shots. Visualization as in Figure S2.

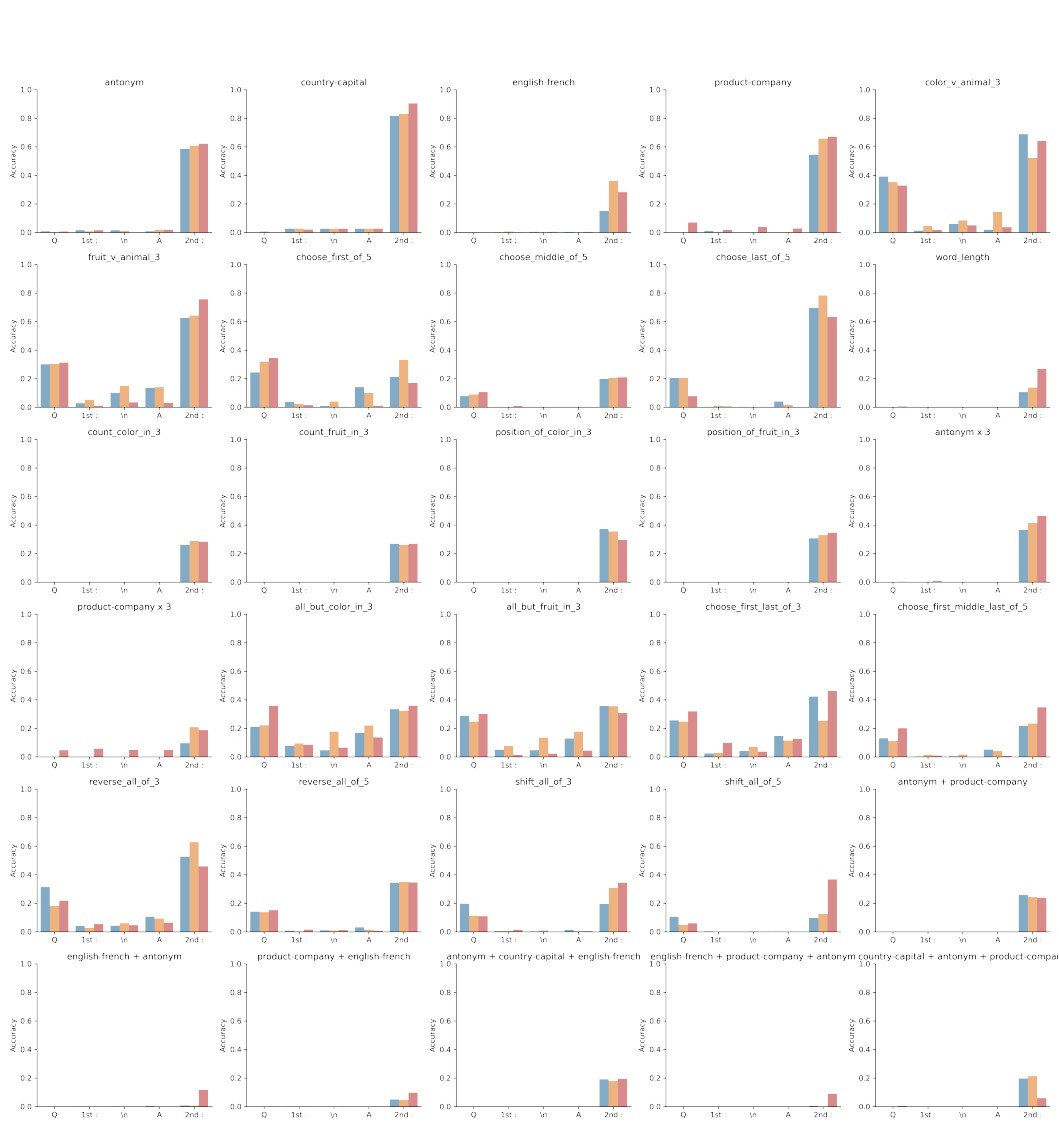

Figure S11: Recontextualized zero-shot accuracy from different format tokens in the prompt. The colors indicate different model sizes: blue=QWEN3_4B, yellow=QWEN3_8B, red=QWEN3_14B.

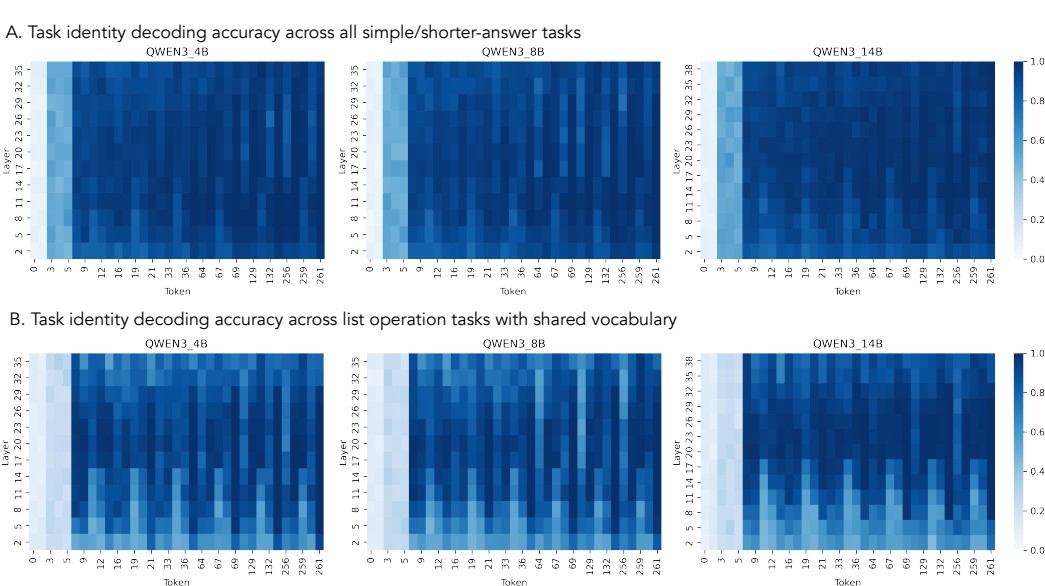

Figure S12: Task identity decoding accroacy across all 14 simple tasks in Table 1 (A) and 9 list operation tasks with shared vocabulary (B). See Figure S4 caption for a complete list of tasks.

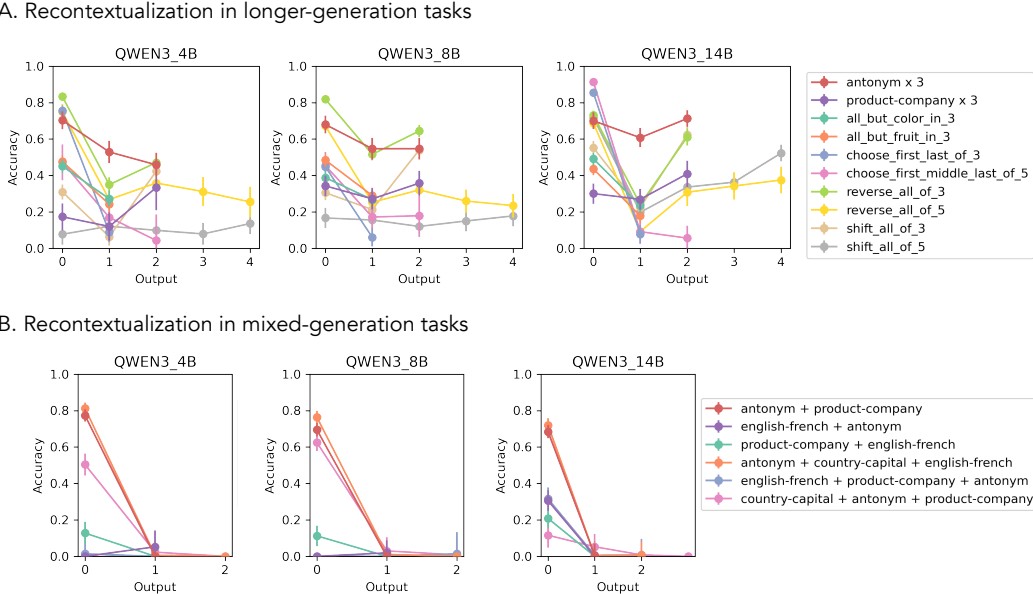

Figure S13: In Qwen3, restored task contexts in longer- and mixed-generation tasks also decay over more output units during generation. Visualization as in Figure 5.

# D TEMPORAL AND SEMANTIC LOCALITY IN FUNCTION VECTORS

We show that the main locality features of transferrable task representations replicate when using function vectors (Todd et al., 2024) as the extraction and recontextualization method. Function vectors are the summed activations of a set of critical attention heads that contribute to in-context task performance. Similar to task vectors, function vectors can transport task semantics across a range of tasks when additively injected onto the last token in a zero-shot prompt.

We build on the procedure outlined in Todd et al. (2024) to extract function vectors from the key last ":" token in 8-shot prompts, with the following exception: we separately measure the critical attention heads contributing to a particular task and extract a per-task function vector. We use the top 16 heads for QWEN3_4B and GEMMA_V3_4B, 32 heads for QWEN3_8B, 48 heads for GEMMA_V3_12B, 64 heads for QWEN3_14B, and 80 heads for GEMMA_V3_27B, roughly corresponding to the number of critical attention heads used in similar-sized models in Todd et al. (2024). To remain comparable with the task vector extraction procedure mentioned above, we similarly sampled at most 50 queries per task to extract function vectors and search for the best intervention layer in zero-shot prompts. We then test the function vector with the best-layer intervention on the remaining queries. In the function vector case, the development set queries are sampled from cases where models correctly generate the target answer in an 8-shot prompt.

Finally, we generalize the extraction of function vectors on other tokens in the prompt to study when this kind of transferrable task representation form. We re-compute the critical attention heads and the best intervention layer per token site. Figures S15 and S16 show replication of the main temporal and semantic scope locality of transferrable function vector representations on Qwen3 models. These results suggest that function vectors and task vectors extract similar kinds of transferrable task representations from models' in-context learning. We unfortunately have yet to be able to induce zero-shot task recontextualization using function vectors on Gemma3 models. We suspect that Gemma3 models' normalization scheme and the increasing magnitude of their residual streams may require scaling up the signals from function vectors when intervening on later layers, but this is a space requiring more investigation.

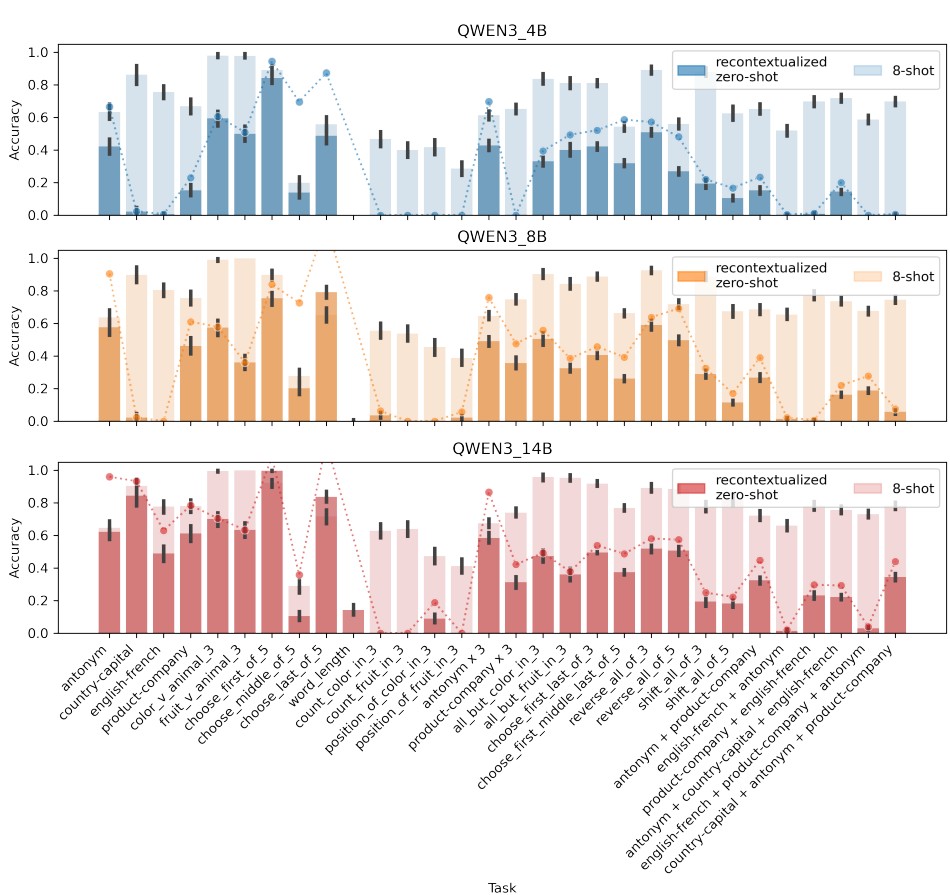

Figure S14: Recontextualized zero-shot task performance using function vectors.

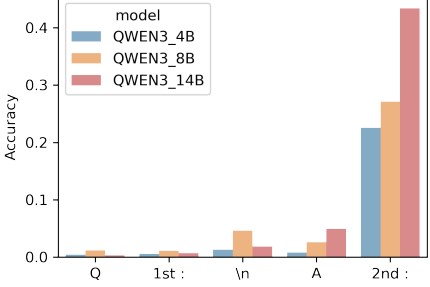

Figure S15: Effective function vectors, like task vectors, tend to only activate on the key colon tokens. The plot shows the zero-shot accuracies when the models are recontextualized with the per-task function vector. Accuracies are averaged across the 14 simple tasks.

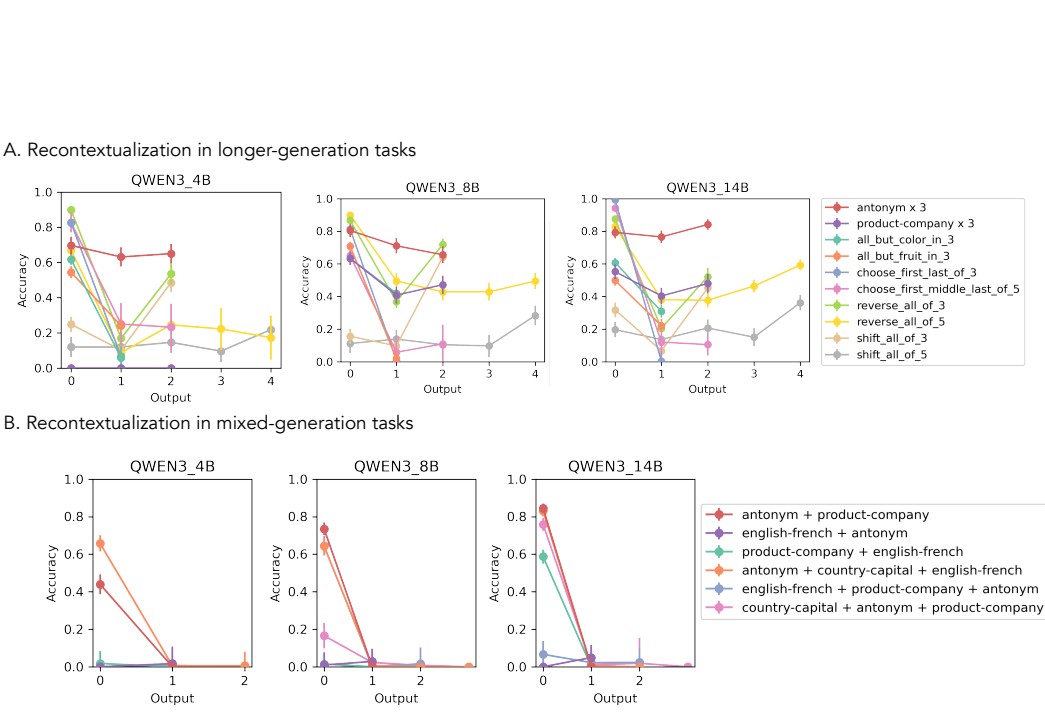

Figure S16: Function vectors extracted from the key colon token similarly capture limited scope locality, especially in cases where the overall task combines semantically independent subtasks. Visualization as in Figure 5.

# E  CROSS-TOKEN ANALYSES

We conduct a set of cross-token transfer and identification experiments to further understand the representational and functional roles of non-key tokens, and the overlap between identifiable and transferable task representations.

**Cross-token task vector transfer** We first test two variants of cross-token task vector intervention. We use representations from non-key tokens in few-shot prompts to intervene the key ":" token in zero-shot prompts (Figure S17). We also tested using representations from the key ":" token in few-shot prompts to intervene different token sites in zero-shot prompts (Figure S18). For each token pair in either experiment, we re-compute the most effective layer of the cross-token intervention.

In both cases, we observe some non-trivial success of task recontextualization. Although there is notable task and model dependence, we find that representations of the ":" token after the "Q" token (prior to query presentation) can be used to help restore task contexts in zero-shot settings. We also find that in Qwen3 models, multiple non-key format tokens can functionally participate in task recontextualization when given effective transferrable task representations. These results suggest a degree of shared task identity representation and transferrable task representation in some tokens. They also indicate that tokens outside of a core circuit for in-context learning (ICL) may nonetheless preserve a useful functional role when given the right information. This could potentially contribute to bypass mechanisms noted in (Cho et al., 2024), where some residual ICL ability remains in the model even when ablating core ICL circuits.

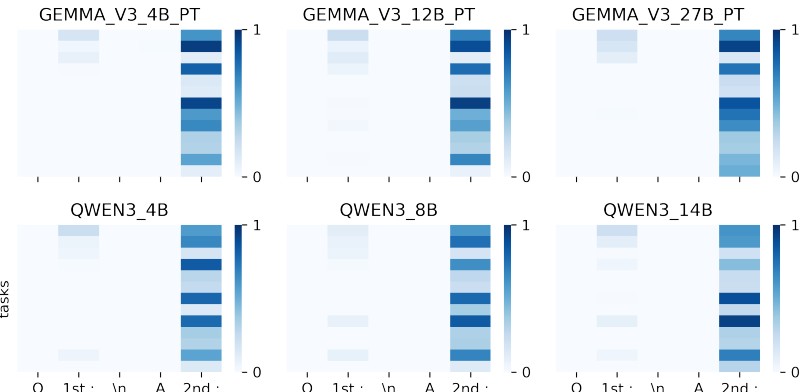

Figure S17: Recontextualized zero-shot accuracy across tasks, using representations from different tokens in 8-shot prompts to intervene the last ":" token in zero-shot prompts. The x-axis indicates the token representations used to intervene zero-shot inference. Each row represents one task. We exclude samples where the correct output is simply the first word of the input query.

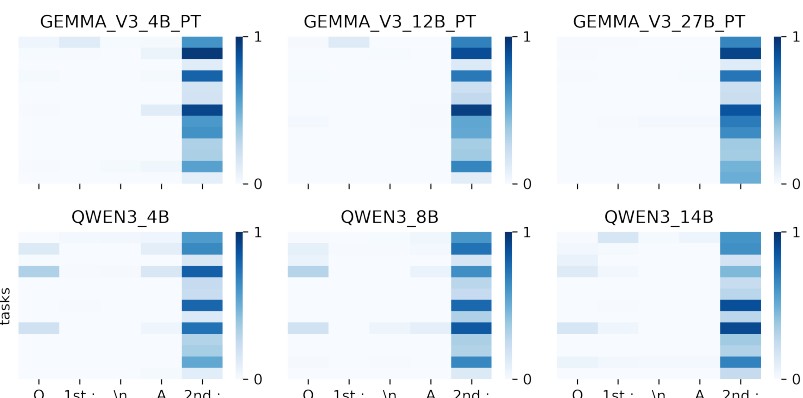

Figure S18: Recontextualized zero-shot accuracy across tasks, using representations from the last ":" token in 8-shot prompts to intervene different tokens in zero-shot prompts. The x-axis indicates the token site intervened during zero-shot inference. Each row represents one task. We exclude samples where the correct output is simply the first word of the input query.

**Analyzing how task identity representations overlap with transferable ones at key tokens:** In this section, we further analyze the relationship between the identifiable and transferable task representations. Specifically, we fit linear classifiers that extract identifiable task representations across all the *non*-effective format tokens—i.e., all format tokens except the 2nd ':' where we find transferable task representations. We also test a version without the first ':' token, given the results above. We then test how well these classifiers generalize to identifying the task representations on the held-out 2nd ':' token representations from a disjoint set of prompts.

We find that there is reliable generalization (Table 4; Figure S19). In fact, we observe perfect generalization for all tasks involving a single output, as well as some of the mixed-generation tasks such as ANTONYM + PRODUCT-COMPANY. However, for many list-operations tasks (e.g. REVERSE_ALL_OF_3), results are somewhat less reliable; but the classifier still achieves robust 75-80% generalization (far above chance performance of 3%, or 6% among these tasks). The performance is comparable with or without the first ':' in the training set for the classifier, suggesting that this generalization is not primarily due to the partially-transferable representations on that token observed above.

Together, these results suggest that *identifiable task representations persist in a relatively consistent way across both tokens that do not produce transferable task representations, and those that do produce transferable task representations*. This suggest that perhaps the identifiable and transferable task representations lie in subspaces that are not fully overlapping.

To understand these results more deeply, we analyzed the relationship between the linear dimensions identified by the regression, and the dimensions of variation in the transferable task representations (Figure S20). We performed PCA over the transferable representations, and identified how strongly the regression features projected onto the top 1-20 principle components. Consistent with the results above, we found that for the same cluster of simple and non-list-operations tasks for which the regressions generalized best, the regression also projected fairly strongly onto the top principle components (with 40%-60% of the identifiable dimension projecting onto the top 20 principle components of transferable representations). By contrast, identifying the other list-operation tasks generally seemed to rely more heavily on components outside the top PCs, with only 10-25% of the identifiable dimension projecting onto the top 20 principle components of the transferable representations. Correspondingly, we find that if we project the transferable representations down to the top 20 principle components, we find strong generalization (99.1%) for the simpler tasks compared to weaker generalization for the other list-operations tasks (34.8%).

Together, these results suggest that *identifiable and transferable task representations lie in partially-overlapping subspaces of the models representations, but the degree of that overlap is modulated by the task type in similar ways to our other findings*. These results reinforce our suggestion that these

distinct types of task representation are distributed differently across the sequence, but shed more light on their interactions and their overlap.

Table 4: Task-identifying regressions generalize from non-transferable tokens to transferable ones.

| Training tokens | Test tokens | Task set | Generalization | Top-20-PCs gen. |
|---|---|---|---|---|
| Q, 1st ':', \n, A | | List-operations | 0.745 | 0.348 |
| | 2nd ':' | Other tasks | 1.0 | 1.0 |
| Q,\n, A | | List-operations | 0.807 | - |
| | | Other tasks | 1.0 | - |

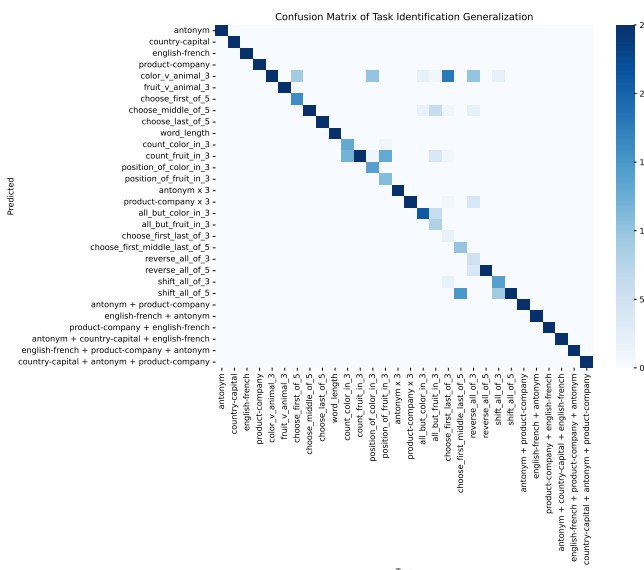

Figure S19: The confusion matrix for the task-identifying regressions (when tested on generalization from non-transferable to transferable representations) shows more details of the errors—while many tasks are identified perfectly, some of the list operation tasks yield more errors. Nevertheless, generalization performance is relatively high even among the list-operations tasks.

Additional details: We performed these experiments in layer 29 of Gemma_V3_27B_PT, which was identified as the mode best layer for transferable task representations. We used 8-shot prompts, and fit the linear classifiers via $\ell^2$-regularized logistic regression. We also fit regressions within only the shared-vocab tasks used in some of the analyses above, and we observe qualitatively similar patterns of results in that setting.

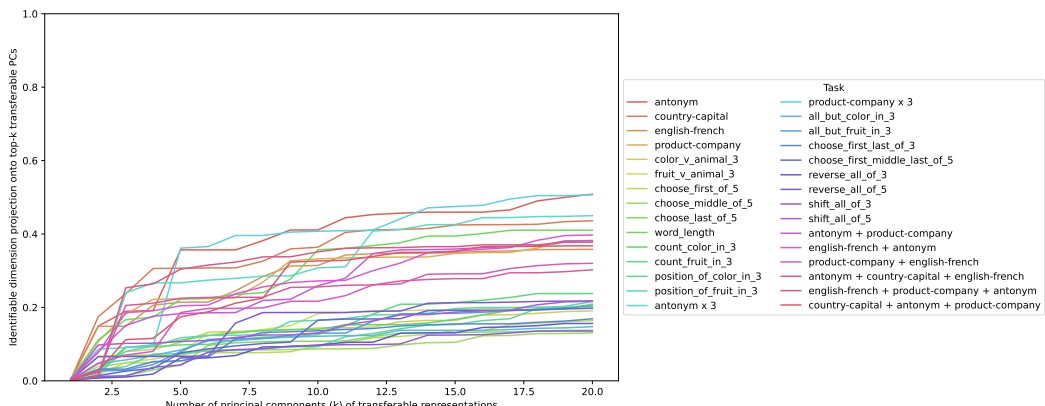

Figure S20: Identifiable task representations partly overlap with the top principle components of transferable task representations, but the degree of this overlap is modulated by the task type.

## F   INSPECTING SUBTASK REPRESENTATIONS IN MIXED-GENERATION TASKS

We showed above that, in mixed-generation tasks, recontextualizing zero-shot inference with task vectors extracted from the key ":" was not enough to support continued generation of subsequent subtask answers. Here, we explore two hypotheses of how subtask representations may be restored through intervention. We take representations from the ":" token in 8-shot prompts and intervene at the "," tokens prior to subtask answers in zero-shot prompts. We compare this intervention to using representations from the matching "," tokens prior to subtask answers in 8-shot prompts to intervene the "," tokens during zero-shot answer generation. Table 5 demonstrates the intervention success when "," tokens prior to subtask generation are intervened with representations from matching prior "," tokens, rather than representations from ":" tokens. The results are consistent with findings from Tikhonov et al. (2025), suggesting that the formation of transferrable subtask representations can be delayed to later format tokens, especially in cases where the subtasks are semantically independent. However, we also note that the restored subtask accuracies are still below that of single-task recontextualization. This may reflect that the full subtask representation requires a combination of task representation in ":" tokens as well as in "," tokens.

Table 5: Interventions on comma tokens in mixed-generation tasks. The table shows the intervened zero-shot accuracy for the output immediately following the intervened "," token versus any remaining output. Results are averaged across tasks.

| Source token | Model | Output after intervened "," | Remaining Output |
|---|---|---|---|
| ":" | Gemma_V3_4B_PT | 0.004 | 0.000 |
| | Gemma_V3_12B_PT | 0.004 | 0.003 |
| | Gemma_V3_27B_PT | 0.003 | 0.014 |
| | Qwen3_4B | 0.003 | 0.000 |
| | Qwen3_8B | 0.001 | 0.000 |
| | Qwen3_14B | 0.004 | 0.009 |
| matched "," | Gemma_V3_4B_PT | 0.455 | 0.002 |
| | Gemma_V3_12B_PT | 0.295 | 0.001 |
| | Gemma_V3_27B_PT | 0.413 | 0.007 |
| | Qwen3_4B | 0.198 | 0.003 |
| | Qwen3_8B | 0.293 | 0.011 |
| | Qwen3_14B | 0.248 | 0.016 |

## G    LLM USAGE

We used GEMINI-2.5-PRO and GEMINI-3-PRO to help polish the writing of this paper. From using GEMINI-2.5-PRO to assist in writing the abstract and the Introduction in an early draft, we adopted the use of "just-in-time" to describe the key findings on the dynamics of transferrable task representations.

