# OpenReview forum: "Just-in-time and distributed task representations in language models"
_ICLR.cc/2026/Conference — Submitted to ICLR 2026_

### Official Review · Reviewer_g6Fb · 2025-10-24

**Soundness:** 2
**Presentation:** 3
**Contribution:** 1
**Rating:** 2
**Confidence:** 3

**Summary:**

This paper investigates the temporal dynamics of transferrable task representations in in-context learning by extracting "task vectors" from few-shot prompts and evaluating their ability to restore task performance in zero-shot settings. The authors claim to discover that transferrable representations activate sporadically at specific tokens (primarily the colon in `Q:A` format) despite task identity being continuously decodable, and that these representations exhibit temporal and scope locality.

**Strengths:**

- Comprehensive experimental design spanning 30 tasks across multiple and recent model families (Gemma V3, Qwen3) and sizes
- Clear distinction between task identity decodability and transferrable task representations, operationalized through different experimental protocols
- Extensive ablations examining task vectors across different numbers of examples ($k$-shot) and different token positions
- Honest acknowledgment of methodological limitations in the discussion section
- Well-designed progression from simple to longer-generation and mixed-generation tasks to probe scope limitations

**Weaknesses:**

**Lack of novelty beyond prior work:** The paper's central methodology—extracting task vectors from the last token position before answer generation—directly replicates Hendel et al. [1]. The finding that this extraction works at the colon token in `Q:A` format is unsurprising given that Hendel et al. demonstrated the same for separator tokens ($\rightarrow$), and Todd et al. [2] further established attention heads as carriers of task information. The "sporadic activation" finding reduces to confirming that task vectors work at positions already identified in prior work and fail elsewhere, which is not a novel insight about representation dynamics.

---
**Overstated claims about "just-in-time" dynamics:** The claim that representations form in a "just-in-time" manner is misleading in my opnion. The paper only probes 4-5 format tokens (`Q`, `:`, `A`, `\n`) rather than systematically examining all token positions throughout the sequence. This sparse sampling cannot support strong claims about temporal dynamics. The paper presents the observation that only the colon token yields transferrable representations as evidence of sporadic activation, but this simply confirms the extraction position from prior work [1-2] rather than revealing dynamic formation processes.

---

**Evidence accrual is trivial:** The finding that task vectors extracted from higher-shot prompts transfer better (Figure 3A) is nearly repetitive--models perform better with more examples, so naturally extracted representations reflect this improvement. This doesn't constitute a novel insight into representation dynamics. The paper doesn't explain why 2 of 14 tasks (COUNT_COLOR_IN_3, COUNT_FRUIT_IN_3) don't show evidence accrual despite behavioral improvement, leaving the generality of findings unclear.

---

**Scope locality cofounded by method:** The decay of recontextualized performance over longer generation (Figure 5) is acknowledged by the authors (page 9) to potentially result from activation patching destroying earlier task state information. Alternative methods using additive injection [2] might not show this decay. Yet this limitation is relegated to a brief discussion rather than being prominently addressed as a methodological confound. The finding is presented as an insight about task representation when it may primarily reflect intervention method limitations.

---

**Missing critical related work:** The paper misses citation of Saglam et al. [3] which proposes _Learnable Task Vectors_ computed as weighted sums of attention heads with layer-specific, causally optimized parameters. This work can be seen the extension of Todd et al., and the omission is problematic given the conceptual overlap.

---

**No discussion of use cases or applications:** The paper provides no discussion of how their findings about representation dynamics could be applied practically. What are the implications, e.g., for prompt engineering, model steering, interpretability, or ICL improvement? Without connecting the observations to actionable insights or applications, the work remains a descriptive study with limited practical utility.

---
**References**

[1] Hendel et al. _In-context learning creates task vectors._ In Findings of the Association for Computational Linguistics: EMNLP 2023.

[2] Todd et al. _Function vectors in large language models._ In ICLR 2024.

[3] Saglam et al. _Learning task representations from in-context learning._ In Findings of the Association for Computational Linguistics: ACL 2025.

**Questions:**

Could the authors clarify how their findings extend beyond prior work establishing that task vectors are optimally extracted from separator tokens  [1-2]? Specifically, what novel insights about ICL mechanisms emerge from observing that the colon token works while other format tokens do not, given that prior work already identified separator tokens as the key extraction point?

---

> ### Author Response · Authors · 2025-11-25
>
> Dear reviewer g6Fb,
>
> Thank you for your time and feedback. We are glad that you find our work comprehensive, and the distinction between representation types and task types valuable. We appreciate hearing your concerns, and hope the following responses clarify our contributions in this work.
>
> **W1+W2: Explicit test of different task representations in different tokens**
>
> We very much agree that the successful task transfer induced by the key “:” token is a direct replication of Hendel et al. (2023)’s. We emphasize the following novel contributions:
> - A key goal of our work is to study distinct types of task representations formed in different tokens: we find consistent representation of task identity even in tokens that do not reliably produce transferable task representations. This comparison was not explored in prior work. We have made this goal more explicit in our revision.
> - Neither Hendel et al. (2023) or Todd et al. (2024) (or other work, to our best knowledge) explicitly investigated whether transferable representations form in other tokens in the context. We believe that directly showing the contrast between key tokens and non-key tokens is valuable, and helps to establish the **scarcity** of this type of task representation during in-context learning.
>
> Our **new cross-token analyses** further reveal novel representational and functional roles of these non-key tokens. As discussed in the *updated section 4.2 and Appendix E*, in some cases, representations from other tokens non-trivially restore task contexts on the “:” token. Non-key tokens also serve as effective sites for task recontextualization when given transferable representations like task vectors extracted from the “:” token. We also find that task identity can be consistently identified on the key “:” token even when using classifiers trained only on other format tokens. There is only partial overlap between the subspaces that carry task identity information and the principle dimensions along which transferable task representations vary, suggesting that identifiable and transferable task representations are conveyed in partly-distinct subspaces. We believe these results offer important clarifications of the nuanced levels of task representations not predictable from prior work.
>
> We also wish to clarify that the four format tokens we examined constitute the majority of format tokens in the prompt we used (except “\n\n” between examples). We acknowledge that our experiments did not include the task-specific query and answer tokens. However, this decision was motivated by the fact that the question-specific information they carry is *necessary* for answering the question correctly; thus copying their representations from one question to another would likely hurt the model’s ability to answer the present question. Thus, we focused our analyses on format tokens that are *consistent* across questions in a task. We argue that understanding the format tokens shared across all sequences and tasks provides a useful characterization of the dynamics of task representations.

---

> ### Author Response · Authors · 2025-11-25
>
> **W3: Evidence accrual in the context of locality**
>
> We agree that our results on the improvement of task vectors over examples are more confirmatory in nature. Our main motivation is to interpret this evidence accrual process in the context of the strong temporal locality, which we feel is a useful clarification of when information aggregation happens within each example. As you note, our results offer some nuance in inconsistent accrual processes across task types. We also provide some surprising evidence on the recovery rate from task vector transfer compared to ICL. Consider (as you note) that the intervention is highly local (i.e. token and layer-specific) and aggressive (i.e., patching), this demonstrates the rich task information these local token representations can contain.
>
> **W4: Scope locality under additive intervention and more**
>
> We appreciate the call for less restrictive interventions. We took upon your suggestion to **replicate our main findings using function vectors**. Function vectors complement task vectors as the intervention is less aggressive (additive injection) and the extracted representation directly captures contributions from multiple attention heads. As shown in *Appendix D*, we find consistent results on temporal and semantic locality when using function vectors to extract transferable task representations. We also made it more explicit in the paper that the limitations of existing methods may indeed limit the generality of our findings (see *updated Discussion*).
>
> Relatedly, we conducted **more experiments** to confirm where subtask representations form in the mixed-generation tasks. Specifically, some subtask representations are delayed to form at “,” tokens prior to subtask answer generation, but the results point to the possibility that multiple tokens collectively form a combination of a global representation of the overall task and more local, distinct representations of subtasks. These new results are now discussed in the *updated section 4.3 and Appendix F*.
>
> **W5: Related work and applications**
>
> Thank you for pointing us to this relevant work and for the call for more discussions. We have added discussion of this work in the revised paper and **added a new paragraph** on the implications of our work for more practical settings.

---

### Official Review · Reviewer_r7d4 · 2025-10-26

**Soundness:** 3
**Presentation:** 4
**Contribution:** 2
**Rating:** 4
**Confidence:** 3

**Summary:**

This paper studies transferable task representations in LLMs—vector representations that can restore task context in another forward pass of the same model. Using Gemma-V3 and Qwen3 families and multiple ICL datasets, the authors extract task vectors, run patching (recontextualization) experiments, and compare against few-shot accuracy. They conclude that transferable task representations exhibit strong locality—appearing sparsely at output tokens—whereas task-identity representations are distributed across many tokens. Long-context generation and combinatorial settings further suggest scope-related locality, where recontextualization restores only subtasks. Overall, the work characterizes where task-relevant information resides and when it transfers in LLMs.

**Strengths:**

1. The paper clearly distinguishes two representation types: an inert task-identity signal that persists across the context versus a transferable signal that activates sporadically at key tokens.

2. The long-context and combinatorial experiments are interesting: recontextualization underperforms few-shot learning in these settings, sharpening the boundary between recognizing a task and transferring it across positions.

**Weaknesses:**

1. Terminology and exposition. Several terms are under-defined (e.g., “temporal,” “scope”), and “recontextualization experiments” is non-standard. Adopting established phrasing such as patching experiments would improve clarity and align with prior work.

2. Novelty relative to prior art. Key observations—(i) that effective task vectors are distributed across multiple tokens have been suggested before (Dong et al.) Please spell out the precise differences and contributions relative to these works.

3. Potentially misleading claim in §3.3. The statement that restored task contexts “support longer generation to some extent … suggesting automatic segmentation into multiple tokens” seems overstated.

4. In Fig. 5, some tasks (e.g., antonym×3, reverse×3 in panel A) show nearly uniform performance under recontextualization.
There is no direct evidence that mixed-generation tasks are decomposed into multiple distinct token-level representations. Stronger analyses (e.g., multi-site patching or ablations across heads/layers) would be needed to support this claim.

**Questions:**

I would increase my score if either (1) or (2) is convincingly addressed—and especially if you demonstrate a more advanced task-vector patching setup (e.g., multi-vector and multi-head/site).

1. Alternative extraction methods :  Can you compare against task-vector extraction methods beyond Hendel et al.? Prior work reports that a single task vector can be insufficient and proposes multi-vector approaches (Tikhonov et al.).

2. Multi-head / multi-site patching :  Some results indicate that single-head, single-site patching is not enough and that aggregating across heads/layers improves recontextualization (Dong et al.). Can you run—or at least discuss—multi-head / multi-site patching?

3. Model class generality : Do these findings (e.g., sparse transferability) extend beyond Transformers to non-Transformer LMs such as RNNs?

References :
1. Hendel et al., "In-Context Learning Creates Task Vectors"
2. Tikhonov et al., “One Task Vector is not Enough: A Large-Scale Study for In-Context Learning.”
3. Dong et al., “Understanding Task Vectors in In-Context Learning: Emergence, Functionality, and Limitations.”

---

> ### Author Response · Authors · 2025-11-25
>
> Dear reviewer r7d4:
>
> Thank you for your time and feedback. We are glad to hear that the clear distinction between the two task representations was valuable, and that the longer and mixed-generation tasks bring interesting learnings. We hope the response below helps address your concerns.
>
> **W1: Terminology use**
>
> Thank you for raising this concern. We were motivated by a high-level characterization of the model’s state of representing tasks that is not captured in more methodology-oriented terms like “patching”, but we fully recognize the need to better integrate with existing terminology. In our revision, we **added a new paragraph** (end of page 4) to provide definitions of the more cognitively-motivated terms we use, and also adopted existing terms in more places to better align with prior work.
>
> **W2: Unique contribution beyond prior work**
>
> Thank you for pointing us to these works. We find it promising that our results are consistent with Dong et al. (2025)’s and Tikhonov et al. (2025)’s. We believe our work extends beyond their findings in the following ways (also **updated in Related Work/Discussion**):
> - Dong et al. (2025) focuses more on how task information aggregates over examples. Our results unpack the task information that exists both *within* each example in different individual tokens and *across* examples.
> - As you noted, we specifically contrast when task vectors form vs. when task identity representations form. The persistent identity representation was not a subject of study in either work. We also **added a set of cross-token analyses** to further understand the coexistence of these two representation types (discussed more below in R2).
> - Although both works noted the distributedness of task representations, our results add important new understandings. Dong et al. (2025)’s multi-task-vector method applies specifically to multiple instances of key separator tokens in few-shot but not zero-shot settings. We show, however, that transferable representations like task vectors can in fact be activated on non-key tokens in certain cases (discussed more in R2 below). Tikhonov et al. (2025) finds distributed task representations in multi-output tasks, but we contrast different types of longer and mixed-generation tasks, which offer distinct nuances and complications (discussed more next).
>
> **W3&W4: Distinct learnings from longer- vs. mixed-generation tasks**
>
> Thank you for helping us recognize that our presentation of the learnings from these two task types can be improved. As you noted, single task vectors can support continued generation in certain cases. This difference between different longer- and mixed-generation tasks suggests that local transferrable representations can sometimes capture global, composite task information, but at other times reflect sharper semantic boundaries. We believe this is an important nuance not captured in Dong et al. (2025) and Tikhonov et al. (2025).
>
> We have refined the discussion around this point (see *updated section 4.3 and Discussion on page 9*) to communicate that the formation of subtask representations likely relates to the semantic independence between subtasks. We also took upon your suggestion to **add more analyses** on how task representations are decomposed into distinct token-level representations (see R3 below).

---

> > ### Author Response · Authors · 2025-11-25
> >
> > We have added three sets of additional experiments to strengthen our paper and offer additional evidence for the questions you raised.
> >
> > **R1: Function vector as an alternative transferable task representation**
> >
> > We experimented with function vectors (Todd et al., 2024) as an alternative method to extract transferable task representations. Function vectors complement task vectors as the intervention is less aggressive (additive injection) and the extracted representation directly captures contributions from multiple attention heads. We show successful replication of our main results, including temporal locality (transferrable task representations exist mostly in key tokens), semantic locality (distinct subtask representations in compositional tasks), and similar differences between longer- and mixed-generation task types. This is now discussed in *Appendix D* and referenced in the main text.
> >
> > **R2: Better understand how both task representations coexist via cross-token analyses**
> >
> > We *added a set of cross-token analyses* to strengthen the comparison between transferable vs. task identity representations. Specifically, we patched task vectors from the key “:” token in few-shot prompts onto non-key tokens in zero-shot prompts, as well as use representations of non-key tokens in few-shot prompts to restore task contexts on the key “:” token in zero-shot prompts. In both interventions, non-key tokens appear to non-trivially restore task contexts, but with substantially lower performance than in the “:” to “:” transfer experiments. This potentially suggests that *some* transferrable task information might overlap with the dominant task identity signal in non-key tokens, and these non-key token sites can also contribute to task prediction when given the right information. We further support these arguments by demonstrating that task identity can be consistently identified on key tokens even when using classifiers trained only on other format tokens, and that there is only partial overlap between the subspaces that carry task identity information and the principle dimensions along which transferable task representations vary. These analyses are now *discussed in a new paragraph on page 8 before section 4.3 and in Appendix E*.
> >
> > **R3: Evidence for multi-site representation in mixed-generation tasks**
> >
> > We share your interest in more fully understanding when effective subtask information aggregates in our mixed-generation tasks. Our *new intervention experiments* test two hypotheses: whether interventions at “,” before subtask generation could restore subtask accuracy via 1) re-injecting the task vector representation from “:” onto “,” tokens or 2) patching representations from matching “,” tokens in prior examples. We find that the latter intervention significantly restored subtask accuracy (from 0% to 20%~40%, depending on the model). However, the reinstated subtask accuracies also fall below their single-task intervention accuracy. Combined with the observation that single-token interventions on “:” in many multi-output tasks could support longer generation, these results point to the possibility that models form a combination of global and more local subtask representations across different tokens. This is now discussed in the *updated section 4.3 and Appendix F*.

---

### Official Review · Reviewer_Krui · 2025-10-27

**Soundness:** 3
**Presentation:** 3
**Contribution:** 1
**Rating:** 2
**Confidence:** 4

**Summary:**

The paper investigate how the task vectors in ICL change across context. On one hand, the task information is represented and can be decoded in many tokens after the first few tokens, showing a continuous pattern along the sequence dimension. On the other hand, when the representions are transferred onto a zeroshot promt to recontextualize and elicit performaing the task, only a few key tokens’ representation can successfully achieve this, showing a sporadic pattern. The paper has done comprehensive experiments to demonstrate this phenomenon and refers this as just-in-time computation. Moreover, the authors also explore transferrability on ICL tasks where the model needs to generate multiple words. They find mixed tasks tend to be represented separately instead of as a combined task.

**Strengths:**

- The paper is well written and the presentation is clear.
- The paper studies longer generation tasks, where there are multiple output words. This is not well-studied by previous research.
- The experiments are comprehensive and the details are handled very well, which makes the results reliable.

**Weaknesses:**

The paper is mainly about observing an interesting phenomenon, however, I find this phenomenon (mainly for temporal locality) seem to be explained by recent works on circuit analysis for ICL:  https://arxiv.org/abs/2410.04468 and  https://arxiv.org/abs/2504.00132 . These works provide deeper understanding about underlying mechanisms behind ICL, and based on their findings, the sequence dimension locality phenomenon emphasized by this paper is somewhat expected.

This paper finds that the task is represented in most tokens throughout the context (after the first few tokens), but only the representations on key tokens (the colon tokens) are transferrable. Transferrable means, when patched onto the zeroshot prompt the model can predict the correct answer.  For non-key tokens, e.g., “A”, the non-transferrability is demonstrated by both (1) patching representation corresponding to“A” (given enough context) to the same layer’s representation corresponding to last colon token of a zeroshot prompt and (2) patching from “A” (given enough context) to the same token “A” in zeroshot prompt. The authors find in both cases the patching does not work, the accuracy is almost zero. This can be explained in terms of the underlying mechanism of ICL. For (1), the given context asks the model to predict colon after “A”, so when patched to the last position the model would follow this context. Because the whole residual stream is patched, not only the task information is transferred, but also template information. If patching in very early layers, both information is missing and the model still cannot do the job. For (2), the last token is colon, so the model would look for occurrences of colon in previous context to aggregate task information. After patching, the task information is only on token “A”, so it is not used or moved to the last position (the colon). See the findings in the mentioned papers. Therefore, the non-transferrability probably results from patching without knowing the model’s circuit.

Given the huge volume of papers these days, it is understandable that the authors do not know some of them. I also understand this can be disappointing. It would be interesting to see some further experiments though. For example, can we identity a task information subspace and only do patching on this subspace to avoid carrying other undesirable information? More specifically, the representation of “A” combines task information, the current token “A”, the information that requests model to follow the template and predict colon. Can we only patch the task information to the last colon of the zeroshot promt? It would be insteresting to identify such subspaces which are not well-studied by previous research.

**Questions:**

- Question:
   - In Figure 5b, on the right, the recontextualization accuracy for output units after the immediate one is almost zero, but why is the accuracy for recontextualization on the left is non-zero? Does that mean the accuracy on the left is computed by averaging the accuracy for each output unit instead of considering each prediction as correct only when it’s correct on all output unit?

- Suggestions:
   - See last paragraph of weaknesses
   - Maybe can try expand the circuit analysis to multi-output setting, which will probably show different mechanisms for the two types of multi-output tasks, longer- and mixed-generation tasks.

---

> ### Author Response · Authors · 2025-11-25
>
> Dear reviewer Krui,
>
> Thank you for your detailed feedback, and for the recognition that our work is well-written and provides comprehensive and reliable results. We very much appreciate the pointers to the circuit-level analysis. In our revision, we added discussion on how our findings provide new learnings and a useful unification of different findings from these circuit analyses, which we also summarize below.
>
> **1. Unique contributions beyond circuit analysis for ICL**
>
> As you noted, our results are consistent with circuit-level findings from Cho et al. (2025) and Bakalova et al. (2025), even when using different methodology to probe task-relevant information. We believe this is an encouraging sign. We want to highlight that our work extends beyond them in several ways (also **updated in Related Work/Discussion**):
> - One of our key motivations is to compare different types of task representations. The rich task identity signal across all format tokens in the prompt is not a subject of study in these works, and not directly evident from their findings.
> - The circuit analyses show that demonstration information per-example aggregates onto key tokens. We additionally show that this aggregation tends to be highly token-exclusive. Our new cross-token analyses (discussed more below in #2) offer additional nuances not predicted from the circuit findings: that non-key tokens seemingly outside of a prediction circuit may also contribute to ICL if given transferrable task representations.
> - Our “activate and fade” framing provides a useful macro-level unification of a few different circuit-level observations from these studies. For example, early separator tokens like “:” do not participate in prediction circuits (Bakalova et al., 2025) as the transferrable task representations they once contain are deactivated and reconstructed later. The contextualization subcircuit observed in Bakalova et al. (2025) results in a convergence of task representation over examples, such that later demonstrations contribute more to the test query as seen in Cho et al. (2025).
> - As discussed below in #3, we show the temporal locality phenomena on a broad range of tasks including but not limited to classification tasks (as in Cho et al., 2025) and tasks that models excel at (as in Bakalova et al., 2025).
>
> **2. New cross-token transfer experiments**
>
> We wish to clarify a small misunderstanding: our submitted version had *not* shown non-transferability from non-key tokens (e.g., “A”) onto a different token (e.g., the key “:” token before answer generation). However, we took the opportunity to test out these kinds of cross-token representation transfer, which revealed further insights. These **new experiments** include:
> - using representations from non-key tokens to restore contexts in the second key “:” token in zero-shot prompts
> - taking representations extracted from the key “:” token and patch onto other tokens in zero-shot prompts
>
> In both cases, we observe non-trivial restoration of task contexts in zero-shot settings that enable the model to perform the target task. There is both model and task dependency, but these non-trivial representations and functional roles of the non-key tokens are not predicted by the circuit mechanisms. They may also contribute to the bypass mechanisms noted by Cho et al. (2025), where some residual ICL ability remains in the model even when ablating the core circuits.
>
> Additionally, we tested how task identity regressions generalize across token types, and find that task identity representations and transferrable task representations co-exist in partially-overlapping subspaces of models’ representations, but this overlap is task-dependent. *This and the above cross-token analyses are now discussed on page 8 before section 4.3 and in Appendix E*.

---

> ### Author Response · Authors · 2025-11-25
>
> **3. Task representations in longer- and mixed-generation tasks**
>
> As you noted, the longer- and mixed-generation tasks in our study provide new evidence on the different kinds of task aggregation that may occur even within a single example. The difference between longer-generation tasks (e.g, repeating tasks, list level tasks) and mixed-generation tasks (distinct subtasks) reveal that models form both global, “mixed-task” representations and local, subtask representations at different tokens.
>
> In our revision, we provide **additional experiments** to clarify where subtask representations aggregate in the mixed-generation tasks. As discussed in the *updated section 4.3 on page 9*, we confirm that representations from the “,” prior to subtask answers can serve as transferable subtask representations and partially restore subtask inference in zero-shot. However, importantly, “,” tokens do not appear to contain sufficient subtask information to restore the full subtask inference ability. Combined with the sustained effect when intervening “:” in longer-generation tasks, these results offer some evidence that different tokens may capture different levels of global and subtask representations.
>
> **4. Accuracies in Figure 5b**
>
> Thank you for capturing this potential confusion. We have now clarified in the Figure caption that the left bar plots depict the average accuracy across all output units, and the line plots depict separate accuracies for each output.

---

### Official Review · Reviewer_zaTd · 2025-10-29

**Soundness:** 3
**Presentation:** 3
**Contribution:** 3
**Rating:** 8
**Confidence:** 3

**Summary:**

This paper investigates how and when large language models (LLMs) form task representations during in-context learning (ICL). The authors focus on “transferrable” task representations—vector representations that can reconstruct task contexts in another model instance even without the full prompt. Through representational and transfer analyses, the study reveals that such representations evolve non-monotonically and sporadically across the input sequence, differing from more stable, high-level task category representations that persist throughout the context. The results show that as more examples are added, the transferrable representations effectively condense task evidence, correlating with improved ICL performance. However, this evidence accumulation process is highly localized—appearing only at specific tokens—indicating that models compute task-specific representations “just in time.” Additionally, these localized representations often correspond to minimal semantic subtasks, while more complex or composite tasks rely on temporally distributed representations. Overall, the paper provides a detailed characterization of the temporal and semantic locality of task representations in LLMs, offering new insights into the mechanisms underlying in-context learning.

**Strengths:**

1.	The paper provides a detailed and original analysis of when and how task representations emerge and evolve during in-context learning, revealing non-monotonic and sporadic development patterns that have not been systematically studied before.

2.	The identification of transferrable representations—those that can restore task contexts across model instances—is a notable conceptual contribution that advances understanding of task encoding in LLMs.

3.	The paper is overall well-written.

**Weaknesses:**

1.	I don’t see any obvious weakness of this paper.

**Questions:**

N/A

---

> ### Author Response · Authors · 2025-11-25
>
> Dear reviewer zaTd,
>
> Thank you for your positive assessment. We are glad that you find our paper well-written and insightful.
> To ensure precise attribution, we would like to clarify that we build upon methods for extracting transferrable representations established in prior work (Hendel et al., 2023; Todd et al., 2024). Our specific contributions lie in identifying and contrasting *different* types of task representations (transferable vs. decodable) and studying the *dynamics* of these representations. We are encouraged to see that the characterizations of the different representational dynamics (locality vs. persistence) resonated with you.
>
> We have taken the opportunity to further strengthen the paper based on feedback from all reviews. In our revision, we have clarified that both types of task representations are the main subjects of our study, and more clearly stated our unique contribution beyond prior work (see highlighted changes in Abstract/Introduction/Related Work/Discussion). We also **added** results from a number of **additional experiments**, including a set of cross-token analyses that help clarify how transferable and task identity information may co-exist in some tokens’ representations (see *updated section 4.2 on page 8 and Appendix E*). These results reveal that, in some cases, non-key tokens can contain useful transferable task information, and even help to recontextualize task contexts when patched with rich transferable task signals.

---

### Author Response · Authors · 2025-11-25
**General response from authors**

We sincerely thank all reviewers for their time and the constructive feedback. We are happy to hear that the reviewers found our experiments **interesting, comprehensive, and reliable**, and that our paper is **well-written and clear**.

These comments helped us identify a few critical points of improvement. We have taken this opportunity to revise our paper to better capture our unique contribution and further our understanding of the representations under study. We summarize the changes here:

- We made more explicit that our key motivation is to compare and contrast different representations that models use to represent in-context tasks (transferrable vs. decodable)
- We expanded the discussion of related works, and clarified how our results both offer a unifying story of multiple prior findings and extend the findings with important nuances and complications
- We conducted **three sets of new experiments** to strengthen our core contribution. We 1) replicate the locality phenomena using function vectors (additive and sensitive to multi-head computation), 2) better disentangle transferable vs. identifiable task information in token representations through cross-token analyses, and 3) verify when and where subtask representations form in longer generation tasks.

We have included more details of these changes in the individual replies, and also highlighted specific revisions in the revised paper.

---

### Author Response · Authors · 2025-12-01
**Comment for new AC**

Dear new AC,

Thank you for considering our paper. As the reviewers did not have time to respond before things got locked, we wanted to briefly summarize our perspective on the exchange. The reviewers generally saw some value in our paper, but pointed to past work with some overlapping findings as well as noting additional experiments that could be useful. In our response, we highlighted key novel findings from our paper, as well as showing ways that our findings illustrate additional complexities that wouldn't be anticipated from prior work. Moreover, we ran substantial additional experiments and analyses (e.g. function vectors, cross-token interventions and linear analyses) that address reviewer questions and introduce new insights. We hope that you will agree that these edits have addressed the reviewer's concerns and made the paper a strong contribution to identifying the types and dynamics of task representations during in-context learning.

Sincerely,
The Authors

---

### Meta-Review · Area_Chair_ypr5 · 2026-01-07

**Summary:**

This paper investigates when and how language models form task representations during in-context learning, distinguishing between transferrable task vectors and task identity representations. The authors find that transferrable representations emerge sporadically at specific tokens (hence just-in-time) while task identity persists throughout context, and demonstrate temporal and semantic locality of these representations across 30 tasks and 6 models. Initial reviews are widely split (ranging from 2 to 8), with three reviewers raising concerns about insufficient novelty relative to recent circuit analysis work and methodological confounds, while one reviewer views the work as original and making notable contributions.

**Reviewer Concerns:**

The rebuttal effectively addressed some core methodological concerns through additional experiments. However, critical concerns remain inadequately addressed: (1) Novelty relative to very recent circuit analysis work. The rebuttal offers only generic claims about providing a unifying story without specific demonstration of what is unified or what novel predictions emerge, (2) Sparse token sampling cannot support strong claims about just-in-time temporal dynamics throughout context, (3) No discussion of practical applications for prompt engineering, model steering, or ICL improvement, and (4) presentation concerns.

**Reviewer Scores:**

Maybe two strong rejects may increase to weak reject/borderline, but core disagreement on novelty and significance seem to persist with three reviewers remaining skeptical despite the additional results.

---

### Decision · Program_Chairs · 2026-01-26

Reject